# GEVRM: Goal-Expressive Video Generation Model For Robust Visual Manipulation

Hongyin Zhang[1,2]    Pengxiang Ding[1,2]    Shangke Lyu[2]    Ying Peng[2]    Donglin Wang[2*]

[1] Zhejiang University. [2] Westlake University.
wangdonglin@westlake.edu.cn

## Abstract

With the rapid development of embodied artificial intelligence, significant progress has been made in vision-language-action (VLA) models for general robot decision-making. However, the majority of existing VLAs fail to account for the inevitable external perturbations encountered during deployment. These perturbations introduce unforeseen state information to the VLA, resulting in inaccurate actions and consequently, a significant decline in generalization performance. The classic internal model control (IMC) principle demonstrates that a closed-loop system with an internal model that includes external input signals can accurately track the reference input and effectively offset the disturbance. We propose a novel closed-loop VLA method GEVRM that integrates the IMC principle to enhance the robustness of robot visual manipulation. The text-guided video generation model in GEVRM can generate highly expressive future visual planning goals. Simultaneously, we evaluate perturbations by simulating responses, which are called internal embeddings and optimized through prototype contrastive learning. This allows the model to implicitly infer and distinguish perturbations from the external environment. The proposed GEVRM achieves state-of-the-art performance on both standard and perturbed CALVIN benchmarks and shows significant improvements in realistic robot tasks.

## 1 Introduction

The pursuit of robust and adaptable robotic systems is the cornerstone of embodied general intelligence. Recently, with the successful advancement of large-scale robot data collection (Vuong et al., 2023), universal state representation learning (Li et al., 2023; Du et al., 2024), and expressive policy learning (Brohan et al., 2022; Chi et al., 2023), the research on vision-language-action (VLA) models for robots has made significant progress. The above strategies have been shown to be effective in estimating the robot's state and generating robust actions in a variety of environments, from physical simulators (Mu et al., 2021; Ding et al., 2023) to carefully designed real-world environments. However, these carefully designed environments do not take into account the inevitable external perturbations during deployment, such as fluctuating lighting conditions or video stream noise due to signal transmission problems. When VLA models are deployed in these non-ideal environments, external perturbations will bring unpredictable state information to the robot. This makes VLA produce fragile and unstable actions in inaccurate environmental states, resulting in a significant decrease in its generalization performance. Therefore, enhancing the robustness of VLA models to cope with the inevitable external perturbations when deployed is an ongoing challenge.

In the fields of computer vision (Simard et al., 2003; Cireşan et al., 2011; Ciregan et al., 2012; Chen et al., 2020) and reinforcement learning (Laskin et al., 2020; Hansen et al., 2021; Zheng et al., 2023), image augmentation is a common technique to alleviate the problem of model over-fitting, resist input image perturbations, and enhance model robustness. The idea is to apply task label-invariant transformations to the model's input images. For example, for object recognition tasks, image flipping and rotation do not change the semantic labels. Therefore, this technique has also

---

*Corresponding author.

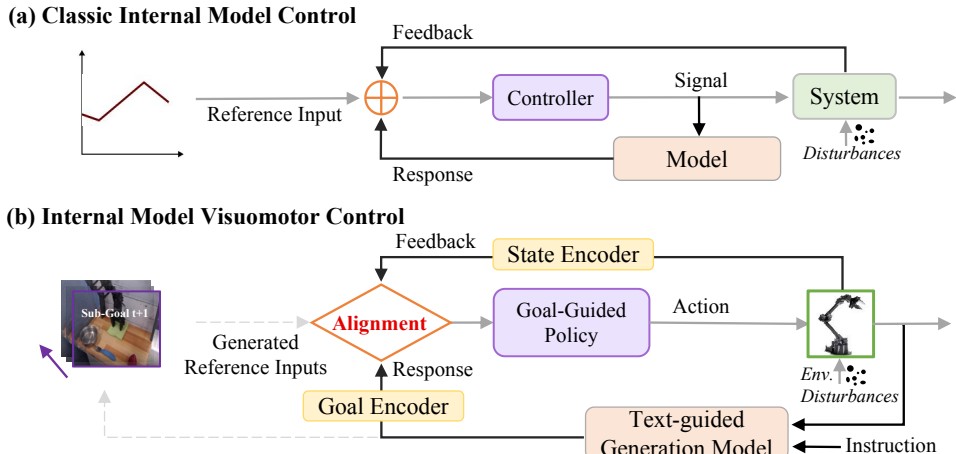

Figure 1: We are inspired by the classical internal model control (**a**) in automation systems. The principle illustrates that a closed-loop system equipped with an internal model that accounts for external input signals can precisely follow the reference input and effectively neutralize the perturbations. In this work, an internal model visuomotor control framework (**b**) is motivated and designed. We leverages a text-guided video model for generating highly expressive visual goal states as reference input, goal-state and current-state internal encoders for modeling responses, and a goal-guided policy for robust action generation.

been applied to robot visual language manipulation tasks. Some previous work has utilized vision as a general medium to develop specific agents that can plan various tasks through imagination and execution (Black et al., 2023; Yang et al., 2023a; Du et al., 2024). These methods involve generative models for predicting future videos or target images, followed by goal-conditioned policies that transform visual plans into actual actions. Image augmentation technology is utilized when training goal-conditioned policies, which to some extent alleviates the policy's over-fitting of specific tasks. However, these models are limited by their generative capabilities, the future goal image (or video) states they generate are not expressive enough, and image augmentation only allows the model to generalize within a narrow task distribution. It lacks strong resilience to environmental perturbations and struggles to produce actions that are consistently effective across diverse task scenarios.

We are inspired by the principle of classical internal model control (IMC) shown in Fig.1 (**a**). The core idea of this principle (Rivera et al., 1986) is that in a closed-loop control system, by building a model inside the controller that can simulate external perturbations and reference inputs, the desired output can be accurately tracked and the perturbations can be effectively offset. That is, it leverages an internal model to replicate the system's behavior and subsequently assess the system's perturbations, thereby augmenting the closed-loop stability. It is widely believed that intelligent mammals also rely on internal models to generate their actions (Nguyen-Tuong & Peters, 2011) and such mechanism is also revealed and supported by behavioral, neurophysiological, and imaging data (Kawato, 1999). More importantly, the integration of the internal model into the robot control system (Emken & Reinkensmeyer, 2005) has been verified to enhance the robustness of the robot motion control. However, the results are limited to specific scenarios and hard to extend to more complex and general tasks, such as visual-language manipulation. How to instantiate the internal model in the VLA framework to improve the robustness of decision actions has not been explored.

To this end, we propose **GEVRM**, a **G**oal-**E**xpressive **V**ideo Generation Model for **R**obust Visual **M**anipulation. As shown in Fig.1 (**b**), to effectively implement the classic IMC principle in the VLA model, some components of our method are adjusted accordingly. **1) Goal generation.** Taking video frames as a universal interface to describe the robot state, we introduce an advanced text-guided video diffusion generation model as a robot behavior planner to generate future goal frames as reference input. To improve the expressiveness of future goal states, we train the visual planner through efficient video spatiotemporal compression and random mask strategies to prioritize the understanding of physical world laws (Kang et al., 2024). **2) State alignment.** We estimate system perturbations by leveraging the simulated responses of the robot. These responses are called internal embeddings and are extracted from the robot state. Since the responses are inherently embedded in the robot's historical observations, the internal embeddings can be optimized through prototypical contrastive learning (Caron et al., 2020; Yarats et al., 2021; Deng et al., 2022) to align the robot's future expressive goal states with its current state. This enables the model to implic-

itly infer and distinguish perturbations from the external environment. **3) Goal-guided policy.** We propose a diffusion policy conditioned on the generated highly expressive goals to better model the multi-modal task distribution of robot manipulation (Chi et al., 2023). This policy and the aforementioned internal embedding are jointly optimized through inverse dynamics and contrastive learning objectives to track highly expressive goals well even in the presence of perturbations. In summary, our contributions are threefold:

- We introduce GEVRM, a novel robust VLA model that incorporates the IMC principle to enhance robot visual manipulation.

- We study how to obtain highly expressive goals with a text-guided video generation model and align state representations through prototypical contrastive learning to resist external perturbations at deployment.

- Extensive experiments verify the effectiveness and advancement of the proposed GEVRM. It significantly outperforms the previous state-of-the-art on the CALVIN benchmark with standard and external perturbations. The expressiveness of the goal states generated in real visual manipulation is significantly improved compared to previous baseline methods.

## 2 RELATED WORK

**Vision-Language-Action models.** With the rise of extensive multi-task robotic datasets (Vuong et al., 2023), the robotics community is increasingly focusing on multi-task execution capabilities. The Vision-Language-Action models (Brohan et al., 2022; Yue et al., 2024) have gained traction for their ability to use language for goal commands, enabling robots to make informed decisions based on visual perceptions. Early studies (Brohan et al., 2022; Wu et al., 2023) utilized cross-modal attention between language and vision, but limited model performance hindered effectiveness. Recently, attention has shifted to large foundational models (Alayrac et al., 2022; Li et al., 2023; Kim et al., 2024), for improved versatility. However, text descriptions often lack detail about environmental states, complicating cross-morphology tasks. As a result, some researches (Du et al., 2024; Ko et al., 2023; Zhou et al., 2024; Black et al., 2023; Ajay et al., 2024; Yang et al., 2023b) now leverage vision as a universal medium, employing generative models to forecast future actions, followed by goal-conditioned policies for execution. UniPi (Du et al., 2024) was one of the first to leverage internet-scale data to train a text-conditioned video generator, using an inverse dynamics model to estimate actions. Similarly, SuSIE (Black et al., 2023) uses an image-editing model to plan high-level sub-goals for low-level controllers, while ADVC (Ko et al., 2023) infers actions from predicted video content with dense correspondences. These efforts aim for a universal state representation but fall short for two reasons. First, existing visual plans experience temporal and spatial inconsistencies due to poor dynamics modeling. We propose a robust video generation model that addresses this issue and enhances action execution. Second, prior work focuses on controlled environments, overlooking the robot's responses to external interference. Our GEVRM method employs contrastive learning for state alignment, effectively simulating responses and resisting disturbances. Together, these elements define our expressive goal representation.

**Internal Model Control framework.** The IMC framework is a widely recognized control strategy that leverages an internal model of the system to predict future behavior and adjust control actions accordingly, making it highly robust against disturbances and model inaccuracies. First introduced by Garcia and Morari (1982), IMC has been applied extensively in both linear and nonlinear process control, offering significant benefits in terms of stability and adaptability (Garcia & Morari, 1982; Rivera et al., 1986; Morari, 1989). Its feedback mechanism allows for real-time adjustments, particularly valuable in dynamic environments such as robotics, where precision is critical. IMC's design has been further explored and refined for multivariable and complex systems, proving its versatility and robustness in various control applications (Skogestad & Postlethwaite, 2005). However, most previous research works are limited to specific control scenarios and are difficult to extend to general visual language manipulation tasks. More recently, inspired by classical closed-loop control systems, a closed-loop visuomotor control framework (Bu et al., 2024) has been proposed that incorporates feedback mechanisms to improve adaptive robot control. Different from these works, we study how to effectively instantiate internal models in the VLA framework to improve the robustness of decision actions.

## 3 PROBLEM FORMULATION

In this work, we investigate how to generate highly expressive goal states and induce robust actions to be resilient to external disturbances. Formally, we study robot trajectory and action generation in a non-Markov decision process framework, which is specified by the following tuple: $M := (\mathcal{X}, \mathcal{A}, \mathcal{G}, \mathcal{T}, \rho_0)$, where $\mathcal{X}$ and $\mathcal{A}$ denote the image state and action spaces, $\mathcal{G}$ represents the language text goal space, $\mathcal{T}(\mathbf{x}_{t+1}|\mathbf{x}_{1:t}, \mathbf{a}_t, \mathbf{g})$ is the transition dynamics, and $\rho_0(\mathbf{x})$ is the initial image state distribution. We aim to generate expressive future image goal states and current actions to be performed given abstract language instructions $g$ and historical image sequence states (i.e., videos) $\tau_{0:t}$ in visual manipulation tasks: $p(a_t, \tau_{t:T}|g, \tau_{0:t})$. The problem is decomposed into two hierarchical levels: 1) Robot behavior planning $--$ given language instructions $g$ and historical video states $x_{0:t}$, infer image goal states $x_{t:T}$; 2) Robot action prediction $--$ given historical and inferred expressive future image goal states $\tau_{0:T}$, predict the current action $a_t$ to be performed. The decoupled process can be expressed as:

$$p_{\Theta}(a_t, \tau_{t:T}|g, \tau_{0:t}) = p_{\phi}(\tau_{t:T}|g, \tau_{0:t})p_{\varphi}(a_t|\tau_{0:T}). \tag{1}$$

This decoupling process greatly reduces the model training's dependence on language, image sequence, and robot action pairs. Specifically, the training of the behavior trajectory planning model $p_{\phi}$ only requires text-video pairs $\mathcal{D}_{\tau,g} = \{(\tau^i, g^i)\}_{i=0}^{I}$ without robot action labels, which can be derived from large-scale video clips with language labels and robot sequence decision data with text annotations on the Internet. The training of $p_{\varphi}$ only requires a small amount of play data $\mathcal{D}_{\tau,a} = \{(\tau^j, a^j)\}_{j=0}^{J}$ without language labels for specific downstream tasks. In the test phase, given a natural language description $g_{test}$ and an initial image state $x_{0,test}$ of a new task, We need to evaluate not only the expressiveness of the future goal states inferred by the model but also the success rate of completing the task under external perturbations.

## 4 METHODOLOGY

Our goal is to build a robust VLA model that incorporates IMC concepts into robotic visuomotor control, as shown in Fig. 2. To set highly expressive goals before execution, we introduce a powerful video generation model as a visual planner (Section 4.1). In Section 4.2, we detail how to align the goal state to evaluate perturbations and show how to induce the generation of robust decision actions. Finally, in Section 4.3, we implement the overall test-time execution pipeline of GEVRM.

### 4.1 ROBOT BEHAVIOR PLANNER

Inspired by the recent success of video generation models (Brooks et al., 2024; Esser et al., 2024), we seek to build a text-guided video diffusion transformer model as a robot behavior planner $P_{\phi}(\tau_{t:T}|g, \tau_{0:t})$ for robotic goal states generation. The planner can faithfully synthesize future goal image frames based on historical video observations and abstract textual task descriptions. Planning via video generation requires a model that can both generate constrained videos starting from the given video and complete the downstream task. Specifically, to obtain highly expressive future goal states, three core aspects need to be considered when designing a robot behavior planner: **1)** video spatio-temporal compression to reduce computing resources, **2)** a random mask mechanism that highlights the understanding of physical laws and object consistency, and **3)** a strong model backbone and an efficient training paradigm.

**Video spatio-temporal compression.** Diffusion transformers (DiT) require massive computational resources to perform complex operations on robot image state sequence data in native pixel space. To alleviate this problem, we first compress the original pixel space with 2D VAE, and then further compress it with 3D VAE to obtain an information-rich low-dimensional dense space. The advantage is that the high computational cost of 3D VAE in the original pixel space is avoided. In fact, after the spatial compression of 2D VAE, there is still considerable temporal correlation between adjacent features. Specifically, during the image state sequences encoding phase, we initially affect spatial dimension reduction by a factor of $8 \times 8$ through the application of 2D VAE (Rombach et al., 2022), and subsequently condense the temporal dimension by a factor of $4\times$ via 3D VAE (Yu et al., 2023). In the image state sequences decoding phase, the temporal dimension is restored prior to the spatial dimension. The 3D VAE incorporates Causal 3D convolutional layers in place of 3D CNNs, ensuring that each frame's output is contingent solely on its antecedent frames (Yu et al., 2023).

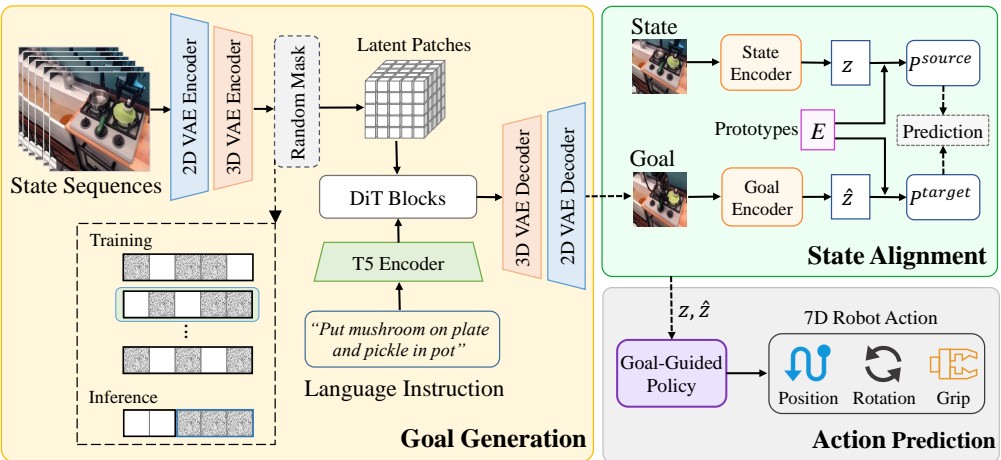

Figure 2: The proposed GEVRM model. First, the *T5* (Raffel et al., 2020) model is utilized to encode language instructions, and 2D and 3D VAE are utilized to compress and restore the original pixel space of the robot image state sequence, followed by the DiT module and random mask mechanism to generate the goal image state. Then, through prototypical contrast learning, the current and goal states are aligned to simulate responses and evaluate perturbations. Finally, the goal-guided policy predicts the 7-dimensional robot decision action.

**Random mask mechanism.** To achieve efficient goal image synthesis, we implement a random mask mechanism (Tay et al., 2022). Precisely, the training process involves the random unmasking of frames, encompassing scenarios such as revealing the initial frame, the first $h$ frames, the final frame, the last $h$ frames, a combination of the initial and last $h$ frames, and arbitrary frames. During the testing phase, we have access to the historical image state but are devoid of future image state. Consequently, in the model's training regimen, the unmasking of the first $h$ frames is assigned the greatest weight, specifically 75%. The remaining unmasking strategies are categorized as supplementary objectives, collectively constituting the remaining 25%. Though the masking mechanism is conceptually straightforward, it enables the robot behavior planner to anticipate subsequent frames based on various temporal snapshots, significantly enhancing the model's comprehension and perception of object dynamics and temporal sequential correlations. For the specific parameter configuration of the random mask, please refer to Appendix Tab. 8.

**Model backbone and training.** Our DiT module is derived from a pre-trained text-guided video generation model (Zheng et al., 2024) and integrates a frozen *T5* encoder (Raffel et al., 2020) to process language instructions (Raffel et al., 2020). Drawing inspiration from the recent advancements in *Stable Diffusion 3* (Esser et al., 2024), we fine-tune robot behavior planner with *Rectified Flow* (Liu et al., 2022), transcending the conventional *DDPM* (Ho et al., 2020). The *Rectified Flow* facilitates the learning of a mapping from noise to real image distribution by solving an ordinary differential equation along straight paths between samples (Liu et al., 2022). This approach has been shown to be a more efficient training paradigm, resulting in a significant reduction in video sampling steps, which in turn significantly increases the model training speed and reduces its inference time.

## 4.2 ROBOT ACTION PREDICTION

The expressive goal state generated by the robot behavior planner is utilized to guide the prediction of decision actions. From the visual goal state $x_{goal}$ and the current visual state $x$ to the final action $a$ output, our goal-guided $\pi_\varphi(a|x_{goal}, x)$ can be divided into the following two parts: **1) State alignment to simulate responses.** Extract informative features from the visual goal state and the current visual state and utilize prototypical contrast learning to align state representations, simulate robot responses, and evaluate disturbances. **2) Goal-guided action prediction.** Decode the goal and current internal compact encoded signals into actions that the robot can robustly perform.

**State alignment to simulate responses.** Within the domain of classical control systems, the IMC framework necessitates the integration of a system's internal model within its controller. This in-

ternal model is capable of offsetting external disturbances and reference inputs, thereby ensuring precise system behavior and reliability. To implement the IMC principle in a learning-based framework, we initially deploy residual networks *ResNet 34* (He et al., 2016) to serve as visual encoders for both the goal and current states. This conversion transforms raw pixel data into an enriched visual representation $f_\psi(x)$, and $f_{\psi'}(x_{goal})$. For the current visual state representation $f_\psi(x)$, the key is how to optimize it to simulate the robotic response for assessing external perturbations, with this response being inherently encoded within the visual goal state. Adhering to the IMC principle, we advocate for the modeling of this process within the latent space $z$ and optimize it through contrastive learning to achieve alignment with the visual goal state.

In the play data $\mathcal{D}_{\tau,a}$, if a pair of $x_{goal}$ and $x$ originates from the same trajectory, they are a positive pair, otherwise they are a negative pair. These pairs are optimized by swapping the assigned tasks (Caron et al., 2020). Specifically, given a sequence of image observations sampled from the play data $\mathcal{D}_{\tau,a}$, we can derive the future goal image $x_{goal}$ from the transition as the target vector, and the current image observation $x$ as the source vector. The source and target vectors are fed to the source and target encoders, respectively, to obtain latent features, which are mapped onto the unit sphere in a high-dimensional space and $\mathcal{L}_2$-normalized:

$$z = \frac{f_\psi(x)}{\|f_\psi(x)\|_2} \text{ , and } \widehat{z} = \frac{f_{\psi'}(x_{goal})}{\|f_{\psi'}(x_{goal})\|_2}. \tag{2}$$

To predict cluster assignment probabilities $p^{source}$ and $p^{target}$ from latent features, we first apply $\mathcal{L}_2$ normalization to the prototypes to obtain a trainable normalized matrix $\mathbf{E} = \{e_n\}_{n=1}^N$, and then take the soft maximum of the dot product of the source or target vectors of all prototypes:

$$p^{source} = \frac{e^{\frac{1}{\delta}ze_n}}{\sum_{n'} e^{\frac{1}{\delta}ze_{n'}}}, \text{ and } p^{target} = \frac{e^{\frac{1}{\delta}\widehat{z}e_n}}{\sum_{n'} e^{\frac{1}{\delta}\widehat{z}e_{n'}}}. \tag{3}$$

Here $\delta$ is the temperature parameter. $p^{source}$ and $p^{target}$ are the predicted probability that the current and goal image observations $x$ and $x_{goal}$ map to individual cluster with index $n$. To obtain the predicted probabilities $\{q_n^{source}\}_{n=1}^N$ and $\{q_n^{target}\}_{n=1}^N$ targets while avoiding trivial solutions, the *Sinkhorn-Knoppal* algorithm (Cuturi, 2013) is applied. Now that we have cluster assignment predictions and targets, the state alignment objective is to maximize the prediction accuracy:

$$\mathcal{J}_\psi = -\mathbb{E}_{x,x_{goal}\sim\mathcal{D}_{a,x}}(q^{source}\ln p^{target} + q^{target}\ln p^{source}). \tag{4}$$

It is worth noting that learning representations to distinguish different instruction and visual representation is a long-standing scientific issue (Pathak et al., 2017), while few studies have explored their ability to simulate robot responses. This capability is not directly accessible in pre-trained visual encoders or policy models learned based only on the current observation (i.e., behavior cloning).

**Goal-guided action prediction.** To keep the model concise, general, and generalizable, we leverage a goal-guided diffusion policy to decode the action output from the state encoding of the simulated response. We only utilize the third-view RGB images from the static camera as input and the action labels as training labels. The robot proprioceptive observation and gripper view images are not applied. The action space of a 7-DoF robot is considered, consisting of the position of the end-effector $a_{EE} \in \mathbb{R}^6$ and the gripper state $a_{gripper} \in \{-1, 1\}$. The goal-guided diffusion policy is a latent variable model using Markov noise and de-noising process, which can be utilized to model the parameterized behavior distribution $\pi_\varphi(a|z,\widehat{z}) = \int \pi_\varphi(a_{0:K}|z,\widehat{z})da_{1:K}$ for the latent variable $\{a_k\}_{k=1}^K$. The forward noise process follows a fixed variance schedule $\{\beta_k\}_{k=1}^K$, which follows the distribution $q(a_k|a_{k-1}) = \mathcal{N}(\sqrt{1-\beta_t}a_{t-1}, \beta_t\mathcal{I})$. Following *DDPM* (Ho et al., 2020), our practical implementation involves directly parameterizing the score network $\pi_\varphi(a_{k-1}|a_k, z, \widehat{z}, k)$ to recover the behavior cloning objective:

$$\mathcal{J}_\varphi = \mathbb{E}_{k\sim\mathbb{U}(1,K),\epsilon\sim\mathbb{N}(0,\mathcal{I}),x,x_{goal},a\sim\mathbb{D}_{x,a}}[\|\epsilon - \pi_\varphi(\sqrt{\widehat{\alpha}_k}a + \sqrt{1-\widehat{\alpha}_k}\epsilon), z, \widehat{z}, k\|_2]. \tag{5}$$

We utilize this objective to train a goal-guided policy and provide it with internal embeddings of the goal and current state. In each policy training iteration, the state encoding is optimized by the state alignment objective, which enables the policy to implicitly infer and distinguish perturbations from the external environment. Therefore, the final optimization objective of the state encoding and goal-guided diffusion policy is:

$$\mathcal{J} = \mathcal{J}_\varphi + \lambda\mathcal{J}_\psi, \tag{6}$$

where $\lambda$ is a temperature parameter. To sample from $\pi_\varphi(a_0|z, \widehat{z})$, a reverse diffusion process where $a_K \sim \mathbb{N}(0, \mathcal{I})$ and $\epsilon \sim \mathbb{N}(0, \mathcal{I})$ is utilized, resampling at each step:

$$a_{k-1} = \frac{1}{\sqrt{\alpha_k}}(a_k - \frac{\beta_t}{\sqrt{1 - \widehat{\alpha}_k}}\pi_\varphi(a_k|z, \widehat{z}, k)) + \sqrt{\beta_t}\epsilon, \text{ for } k = \{K, \cdots, 1\}. \tag{7}$$

## 4.3 Test-time Execution Pipeline of GEVRM

Once both the robot behavior planner $P_\phi$ and the goal-guided policy $\pi_\varphi$ are trained, they can be utilized to solve new manipulation tasks. Given a new scenario $x_{0,test}$ and a new language command $g_{test}$, GEVRM attempts to solve the task by iteratively generating highly expressive goal states and inducing goal-guided policies to achieve these sub-goals. Initially, we sample a set of goals $\{x_{m,goal}\}_{m=0}^M \sim P_\phi(\cdot|x_{0,test}, g_{test})$, where $M$ represents the number of goal state generation. We pass goal state $x_{t,goal}$ and current state $x_t$ through the state encoders over $L_{test}$ time steps to obtain the internal embedding and derive the goal-guided policy $\pi_\varphi$, where $L_{test}$ is the fixed interval number. After $L_{test}$ time steps, we refresh the goal states by sampling from the behavior planner again and repeat the process. The test execution of the algorithm is shown in Algo. 1.

---

**Algorithm 1** GEVRM: Test-time Execution

1: Robot behavior planner $P_\phi$, state encoder $f_\psi$, goal state encoder $f_{\psi'}$, goal-guided policy $\pi_\varphi$, time limit $T$, goal sampling interval $L_{test}$, goal generation number $M$, initial state $x_{0,test}$, language instruction $g_{test}$.
2: $t \leftarrow 0$
3: **while** $t \leq T$ **do**
4:      Sample goals $\{x_{m,goal}\}_{m=t}^{t+M} \sim P_\phi(\cdot|x_{t,test}, g_{test})$        $\triangleright$ Behavior planner generates goals.
5:      **for** $l = 1$ to $L_{test}$ **do**
6:          $z_t \leftarrow \frac{f_\psi(x_{t,test})}{\|f_\psi(x_{t,test})\|_2}$        $\triangleright$ State encoding and $\mathcal{L}_2$ normalization.
7:          $\widehat{z}_l \leftarrow \frac{f_{\psi'}(x_{l,goal})}{\|f_{\psi'}(x_{l,goal})\|_2}$        $\triangleright$ Goal state encoding and $\mathcal{L}_2$ normalization.
8:          Sample action $a_t \sim \pi_\varphi(\cdot|z_t, \widehat{z}_l)$        $\triangleright$ Goal-guided action prediction.
9:          $x_{t+1,test} \leftarrow \text{Env.Step}(a_t)$
10:          $t \leftarrow t + 1$
11:      **end for**
12: **end while**

---

## 5 Experimental Evaluation

In this section, we evaluate the state generation and visual manipulation capabilities of GEVRM. To this end, our experiments aim to investigate the following questions: **1)** Can GEVRM have strong generalization ability to generate expressive goal in various environments? **2)** Does GEVRM exhibit a higher success rate in executing robot tasks compared to the baseline in various environments? **3)** How important are the core components of the GEVRM for achieving robust decision action?

### 5.1 Evaluation on Goal Generation

**Setup.** We utilized two types of datasets (realistic Bridge (Walke et al., 2023) and simulated CALVIN (Mees et al., 2022b)) to evaluate the generalization of goal generation. We train the model on a predefined training set and evaluate the robot goal generation performance on a test set with and without external perturbations. The hyperparameters are shown in Appendix Tab. 8 and Tab. 9.

**Baselines.** To make a fair comparison, we have chosen open-source video generative models: 1) AVDC (Ko et al., 2023), a typical diffusion-style generation model for robotics. 2) GR-1 (Wu et al., 2023), which is an autoregressive-style generation model that takes language instructions, and state sequences as inputs, and predicts robot actions and future images in an end-to-end manner. 3) SuSIE (Black et al., 2023), uses the image editing diffusion model as a high-level planner and proposes intermediate sub-goals that can be achieved by the low-level controller.

**Metrics.** The evaluation metrics employed are the Frechet Inception Distance (FID) (Seitzer, 2020) and the Frechet Video Distance (FVD) (Skorokhodov et al., 2021; Yu et al., 2022), both widely recognized in the domains of image and video generation. We also evaluate the quality of videos

Table 1: Goal generation quality comparison. Our method greatly surpasses the baseline across all metrics. The best results for each task are bolded.

| Benchmark | Algorithms | FID ($\downarrow$) | FVD ($\downarrow$) | LPIPS ($\downarrow$) | SSIM ($\uparrow$) | PSNR ($\uparrow$) |
|---|---|---|---|---|---|---|
| BridgeData | AVDC | $246.45_{\pm 39.08}$ | $22.89_{\pm 4.99}$ | $0.23_{\pm 0.03}$ | $0.73_{\pm 0.05}$ | $18.22_{\pm 2.53}$ |
| BridgeData | SuSIE | $114.79_{\pm 21.38}$ | $-$ | $0.22_{\pm 0.08}$ | $0.71_{\pm 0.07}$ | $16.39_{\pm 2.90}$ |
| BridgeData | GEVRM (Ours) | $\mathbf{35.70}_{\pm 10.77}$ | $\mathbf{4.16}_{\pm 1.35}$ | $\mathbf{0.06}_{\pm 0.03}$ | $\mathbf{0.89}_{\pm 0.04}$ | $\mathbf{22.36}_{\pm 2.75}$ |
| CALVIN | GR-1 | $236.75_{\pm 38.87}$ | $12.83_{\pm 2.60}$ | $0.20_{\pm 0.02}$ | $0.65_{\pm 0.03}$ | $18.59_{\pm 0.95}$ |
| CALVIN | SuSIE | $214.14_{\pm 45.45}$ | $-$ | $0.15_{\pm 0.04}$ | $0.75_{\pm 0.05}$ | $18.12_{\pm 2.29}$ |
| CALVIN | GEVRM (Ours) | $\mathbf{94.47}_{\pm 22.54}$ | $\mathbf{3.80}_{\pm 1.2}$ | $\mathbf{0.09}_{\pm 0.04}$ | $\mathbf{0.80}_{\pm 0.05}$ | $\mathbf{21.10}_{\pm 3.29}$ |

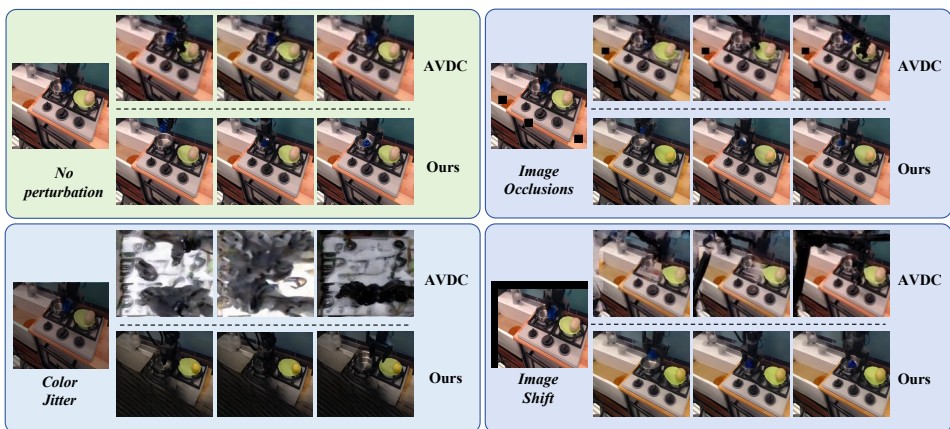

Figure 3: Comparison of goal generation on task "*put blueberry in pot or pan on stove*".

generated by different models on other standard metrics (Bu et al., 2024): Structural Similarity Index (SSIM), Peak Signal-to-Noise Ratio (PSNR), Learned Perceptual Image Patch Similarity (LPIPS).

**Goal generation comparison.** We evaluate the generalization of the goal generation in the unseen environment (Tab. 1). The results show that the performance of our GEVRM model is significantly improved compared to the baseline. The results indicate that GEVRM exhibits enhanced expressive capabilities, effectively modeling the intricate textures and temporal coherence of robotic image sequences. Then, we compare the robustness of goal generation in the perturbed environment (Fig. 3). The baselines struggle with environmental variations, generating severe hallucinations that distort objects and may even completely ruin the scene. In contrast, our method produces fewer hallucinations and can generate expressive goal states following language instructions. This confirms that GEVRM is indeed better able to understand the laws of the physical world and maintain the 3D consistency of objects. More goal generation results are in Appendix Fig. 9~15.

## 5.2 EVALUATION ON ACTION EXECUTION

**Setup.** We conduct experiments on CALVIN, a benchmark for language-conditioned manipulation to evaluate the GEVRM's capabilities in closed-loop action execution. CALVIN consists of four simulated environments (A, B, C, and D), each with a dataset of human-collected play trajectories. We study zero-shot multi-environment training on A, B, and C, and testing on D, varying in table texture, furniture positioning, and color patches. We also test GEVRM's robustness to perturbations (Fig. 4). Details of environmental disturbances are given in Appendix Section A.1. The policy training hyperparameters are shown in Appendix Tab. 10.

**Baselines.** We select the representative baselines to verify the generalization performance on standard unseen environments: 1) UniPi (Du et al., 2024): Recasts decision-making into text-conditioned video generation firstly, enabling the production of predictive video sequences and subsequent extraction of control actions. 2) HiP (Ajay et al., 2024): This model improves upon UniPi by incorporating hierarchical inference to extend long-term planning capabilities. 3) GR-1 (Wu et al.,

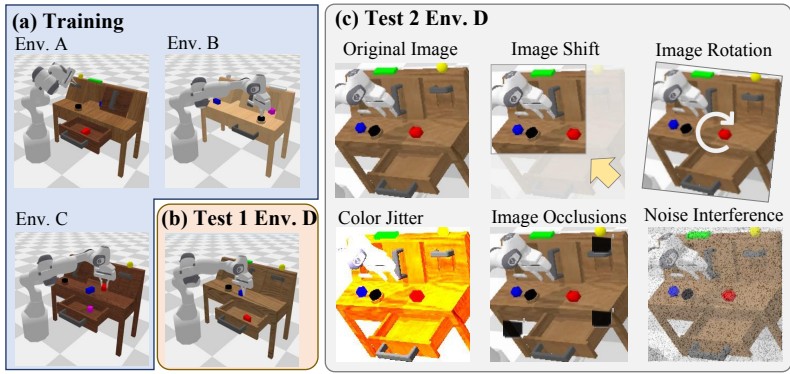

Figure 4: The model is trained only on data collected in environments A, B, and C (**a**), and tested on environment D (**b**). Besides, we apply five perturbations to the image observations of environment D to further test the generalization of the model in more challenging scenarios (**c**).

2023): This model leverages a pre-trained video model to enhance autoregressive action generation. 4) RoboFlamingo (Li et al., 2023), uses a pre-trained VLM for single-step visual language understanding and models sequential history information with an explicit policy head. In addition, in the test environment with external perturbations, we choose the representative baseline SuSIE (Black et al., 2023), because it adopts common data augmentation strategies to cope with perturbations and achieves state-of-the-art results in previous works. We consider third-view RGB images from static cameras as observations, which makes the robot execution more challenging. More comparison with language-conditioned methods are in Appendix Sec. A.2 and Tab. 5.

**Action execution comparison.** We show the success rate of completing each language instruction in the chain in Tab. 2. The model is trained on environments A, B, and C (Fig. 4 (**a**)), and test in D (Fig. 4 (**b**)). Compared with the baseline, the GEVRM has a significant performance improvement. This shows that our method based on the IMC principle has better goal generation ability when facing new environments and induces the robot to predict more general decision actions.

Table 2: Generalization on unseen environments in CALVIN (train A, B, C → test D). *: reproduced version training on third-view images.

| Algorithms | No. of Instructions Chained | | | | |
|---|---|---|---|---|---|
| | 1 | 2 | 3 | 4 | 5 |
| HiP | 0.08 | 0.04 | 0.00 | 0.00 | 0.00 |
| UniPi | 0.56 | 0.16 | 0.08 | 0.08 | 0.04 |
| GR-1* | 0.75 | 0.45 | 0.2 | 0.15 | 0.10 |
| SuSIE | 0.87 | 0.69 | 0.49 | 0.38 | **0.26** |
| GEVRM (Ours) | **0.92** | **0.70** | **0.54** | **0.41** | **0.26** |

Table 3: Generalization on perturbed environments in CALVIN (train A, B, C → perturbed test D).

| Five Perturbed Tasks | Algorithms | No. of Instructions Chained | | | | | Avg. Length (↑) |
|---|---|---|---|---|---|---|---|
| | | 1 | 2 | 3 | 4 | 5 | |
| Average | SuSIE | 0.56 | 0.26 | 0.13 | 0.10 | 0.06 | 1.11 |
| | RoboFlamingo | 0.63 | 0.35 | 0.18 | 0.09 | 0.05 | 1.31 |
| | GR-1 | 0.67 | 0.38 | 0.22 | **0.11** | 0.06 | 1.44 |
| | GEVRM (Ours) | **0.70** | **0.47** | **0.26** | **0.11** | **0.07** | **1.62** |

**Action execution comparison under external perturbations.** To thoroughly evaluate the performance of our proposed GEVRM against the baseline SuSIE, we tested both models across five more challenging scenarios (Fig. 4 (**c**)). The average performance on five perturbed tasks is in Tab. 3, and the specific results are in Appendix Tab. 4. These scenarios were crafted to challenge the models' perception of environmental stimuli and comprehension of physical laws. The results show that GEVRM can well simulate robot response and guide the policy to generate robust decision actions to resist external perturbations. More action execution comparison results are in Appendix Tab. 5. Real-world deployment of GEVRM is in Appendix Sec. A.3.

## 5.3 ABLATION STUDY

In this section, we perform an ablation study to assess the contributions of VAE and state alignment to our method. We evaluate the effects of VAE fine-tuning and state alignment application on

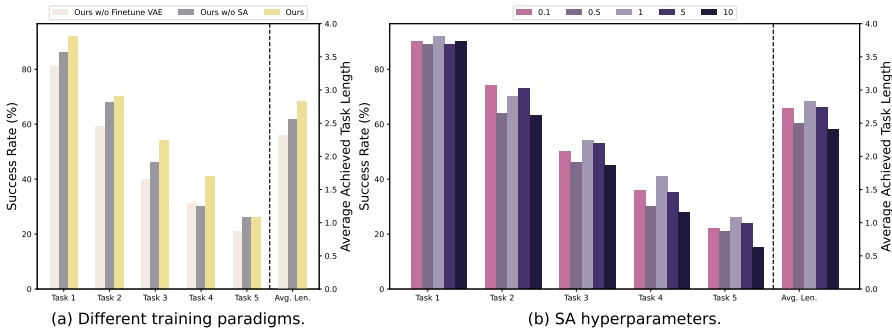

Figure 5: Ablation study on the CALVIN ABC → D. (a) We compare different training paradigms. (b) We examine the impact of different values of the state alignment (SA) hyperparameter $\lambda$.

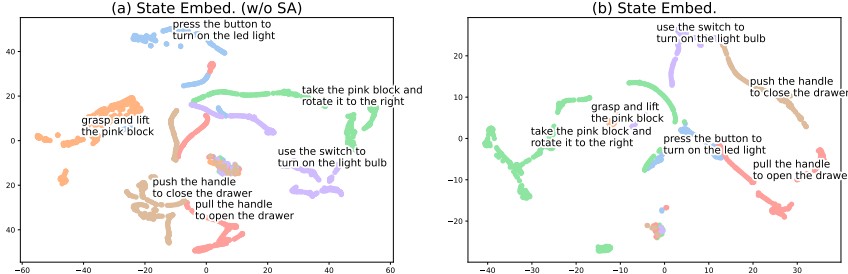

Figure 6: Comparison of state representations. The representations with state alignment (SA) show enhanced cluster centers, class boundaries, and temporal consistency.

model performance across CALVIN environments A, B, and C, focusing on robot behavior planning and goal-guided policy training. Results in Fig. 5 (a) indicate that omitting VAE fine-tuning or state alignment integration significantly degrades model performance on CALVIN Env. D, due to the VAE's pre-training on diverse video data enhancing spatio-temporal consistency and subsequent fine-tuning on robot data aiding in decision-making generalization. State alignment bolsters the policy's visual state representation for better task generalization. Moreover, the hyperparameter $\lambda$, crucial for balancing expert imitation and state alignment in policy training, was tested across five values (Fig. 5 (b)). Performance metrics varied minimally, showing robustness to $\lambda$ adjustments, with $\lambda = 1$ optimal for our method. To illustrate the impact of state alignment on goal-guided representation, we conducted a visual comparison experiment. We utilize *T-SNE* (Van der Maaten & Hinton, 2008) to analyze the latent space representations of current and future image states with and without state alignment in the CALVIN ABC → D "Noise Interference" task, and the results are shown in Fig. 6 and Appendix Fig. 8. Results indicated that state alignment improves clustering and classification by enhancing intra-category cohesion and inter-category separation. Additionally, state alignment ensures temporal consistency in image state sequences, thereby bolstering the policy's environmental and task recognition, and facilitating generalization to novel scenarios. The ablation experiments on the execution efficiency of the goal generation and the goal-guided diffusion policy are in Appendix Tab. 6 and 7, respectively.

## 6 CONCLUSION

The innovation of our method lies in its ability to internalize the classical internal model control principle into the modern VLA framework, thereby enhancing the robot's ability to handle environmental perturbations and maintain performance integrity. In the proposed robust GEVRM model, we leverage video generation models to obtain highly expressive target states. Meanwhile, we effectively align state representations based on prototype contrastive learning to simulate robot responses and evaluate external perturbations. As demonstrated by the GEVRM's state-of-the-art performance in simulated and realistic visual manipulation tasks, it effectively enhances the expressiveness of the goal state and exhibits strong resilience to external perturbations. Therefore, our work greatly expands the reliability and robustness of robotic systems in deployment scenarios and is an important step forward in the field of embodied general intelligence. A promising work is to consider incorporating more general high-quality video generation models into the VLA framework to cope with complex and diverse manipulation tasks of real-world robots.

ACKNOWLEDGMENTS

This work was supported by the National Science and Technology Innovation 2030 - Major Project (Grant No. 2022ZD0208800), and NSFC General Program (Grant No. 62176215).

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

# A    APPENDIX

## A.1    ENVIRONMENT PERTURBATIONS.

**CALVIN Datasets.** We conduct experiments on CALVIN (Mees et al., 2022b), a benchmark for long-horizon, language-conditioned manipulation to evaluate the GEVRM's capabilities in closed-loop action execution. CALVIN consists of four simulated environments (A, B, C, and D), each with a dataset of human-collected play trajectories. Each environment consists of a *Franka Emika Panda* robot arm positioned next to a table with various manipulable objects, including drawers, sliding cabinets, light switches, and colored blocks. Environments are distinguished by their tabletop texture, the positions of furniture objects, and the configuration of colored blocks. We study zero-shot multi-environment training on A, B, and C, and testing on D, varying in table texture, furniture positioning, and color patches.

**Environment perturbations Details.** We list the details of perturbations in Fig 4: **1)** The image state is randomly translated to the upper left, with a maximum translation ratio of 0.1 relative to the image size. **2)** The image state is randomly rotated counterclockwise, with a maximum rotation angle of 30 degrees. **3)** The image state saturation, brightness, contrast, and sharpness are randomly jittered with a maximum random factor of 3. **4)** The image state is randomly occluded with a random number of occlusion blocks ranging from 1 to 3 and a maximum length of 60. **5)** The image state is perturbed with random noise blocks. We follow the evaluation protocol of Mees et al. (Mees et al., 2022b). During the evaluation, the policy is required to complete five chains of language instructions for 360 time steps. Notably, we only consider RGB images from the static camera as observations, which makes CALVIN much more challenging.

## A.2    BASELINE METHOD INTRODUCTIONS.

For a fair comparison, we here select the video generation-based baselines to verify the zero-shot generalization performance on standard unseen environments: These baseline methods include language-conditioned policies that leverage pre-trained visual-language models in various ways: **1) HULC** (Mees et al., 2022a), a model that employs a multi-modal transformer encoder for language-conditioned robotic manipulation, combines self-supervised contrastive learning to align video and language representations and uses hierarchical robotic control learning to tackle complex tasks. **2) MCIL** (Lynch & Sermanet, 2020), a multi-context imitation learning framework, is capable of handling large-scale unlabelled robot demonstration data. MCIL trains a single goal-conditioned policy by mapping various contexts, such as target images, task IDs, and natural language, into a shared latent goal space. **3) MdetrLC** (Kamath et al., 2021), which integrates visual and textual information to perform object detection and multi-modal understanding. MdetrLC uses text query modulation to detect objects within images and demonstrates strong performance in tasks like visual question answering and phrase localization. **4) AugLC** (Pashevich et al., 2019), which optimizes image augmentation strategies to enable Sim2Real policy transfer from simulated environments to real-world scenarios, applies random transformation sequences to enhance synthetic depth images and uses auxiliary task learning to reduce the domain gap between synthetic and real images. **5) LCBC** (Walke et al., 2023), which employs ResNet-34 as the image encoder combined with MUSE language embeddings for robotic decision-making, uses FiLM conditioning to embed language information into the visual encoding, which in turn generates robot actions. **6) UniPi** (Du et al., 2024), which transforms decision-making problems into text-conditioned video generation tasks, produces future video sequences of the target task and extracts control actions from the generated videos. **7) HiP** (Ajay et al., 2024), an extension of the UniPi method, enhances the model's ability to handle long-horizon tasks by introducing hierarchical inference and planning. This approach decomposes tasks into high-level planning and low-level action generation, improving task execution in complex scenarios. **8) SuSIE** (Black et al., 2023), which leverages a pretrained image-editing diffusion model to generate sub-goal images, guides the robot through complex manipulation tasks via language instructions. SuSIE integrates a large-scale internet visual corpus during sub-goal generation and achieves these generated sub-goals through a low-level goal-oriented policy.

*Instruction: Pick up the red cup. SR:0.8*

*Instruction: Put the smaller blue bowl into the red bowl. SR:0.8*

*Instruction: Take the tiger out of the red bowl. SR:0.6*

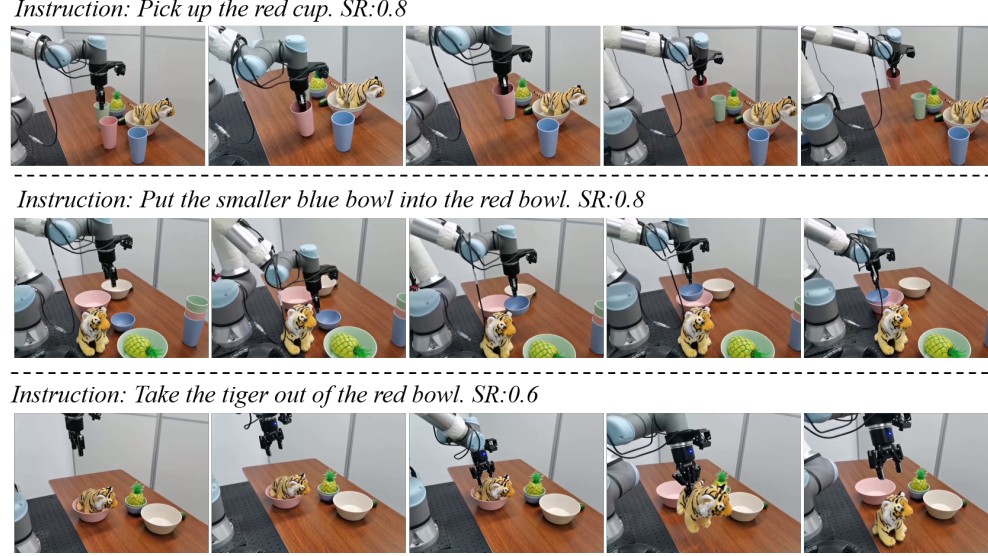

Figure 7: Real-world task results. Our method GEVRM can be effectively deployed in real-world scenarios, such as the picking and placing tasks of cups, bowls and tiger plush toys.

### A.3 REAL-WORLD TASKS.

**Protocol.** To examine the effectiveness of the proposed GEVRM on real-world robotic manipulation tasks, we propose a real-machine deployment protocol. We evaluate GEVRM on a robotic arm UR5 for the pick-and-place tasks of a cup, a bowl, and a tiger plush toy. Specifically, we use a camera to capture third-person images as the observation space (image width 640, height 480), and relative poses and binarized gripper states as the action space (7 dimensions). The total number of collected real-world teleoperation expert trajectories is over 400, with trajectory lengths ranging from 20 to 120 steps and a control frequency of 5Hz.

**Experiments.** We train and evaluate GEVRM under real-world protocols. The VAE and DiT in the behavior planner are trained for 30,000 and 12,000 iterations, respectively, while the goal-guided policy is trained for 100,000 iterations. Other hyperparameters remain the same as in the experiments in CALVIN (and Bridge). Fig. 7 shows the policy execution process of our proposed GEVRM on three types of real-world tasks, indicating that our method can be effectively deployed on real machines. In terms of task success rate (SR), we evaluated each type of task 10 times. The experimental results show that compared with the grasping and placing of cups (or bowls) with regular shapes (success rate of about 0.8), the grasping of tiger plush toys with soft materials and irregular shapes is more challenging (success rate of about 0.6). Further improving GEVRM's perception of real-world scenes and task execution accuracy is an important future work.

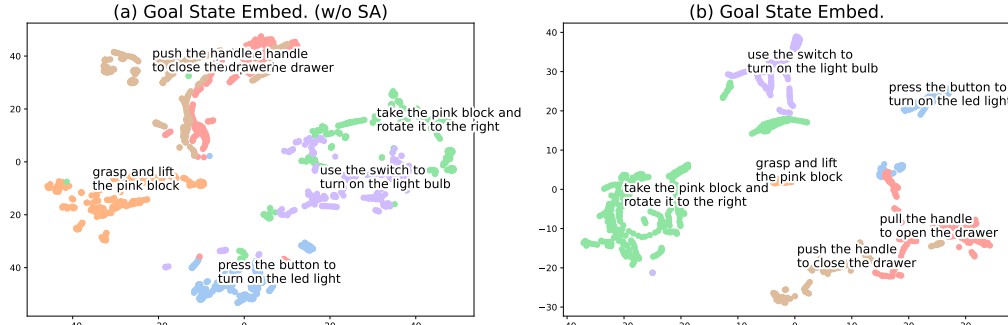

Figure 8: Visual comparison of the latent space representation of the goal image state with and without state alignment (SA). The gaol state representation sequence with SA also has better cluster centers, category boundaries, and temporal consistency.

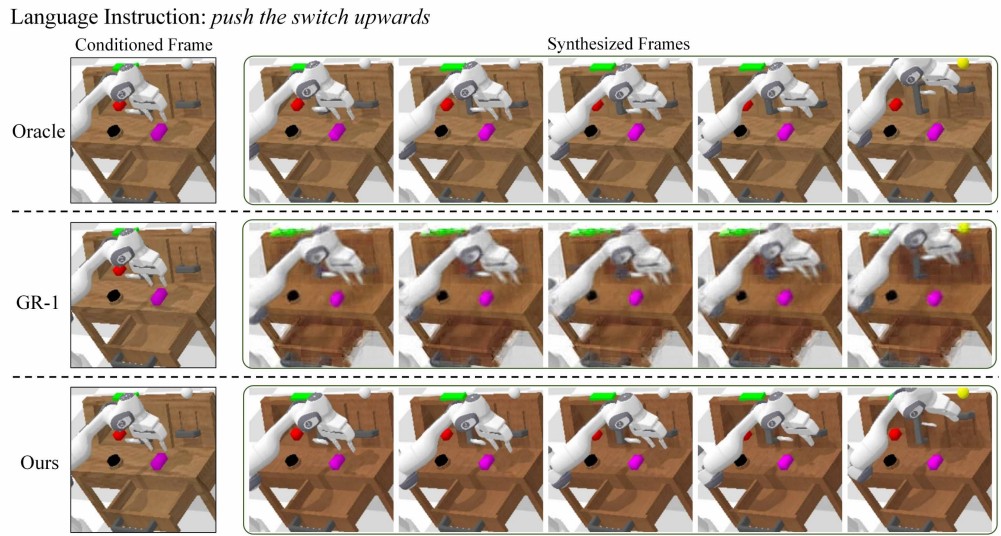

Figure 9: Comparison of goal generation. We visually compare Oracle, GR-1, and our proposed algorithm on the "*push the switch upwards*" task in CALVIN Env. D. Compared with the baseline GR-1, the goal video we generate is more realistic and better restores details.

Language Instruction: *go towards the blue block in the drawer and lift it*

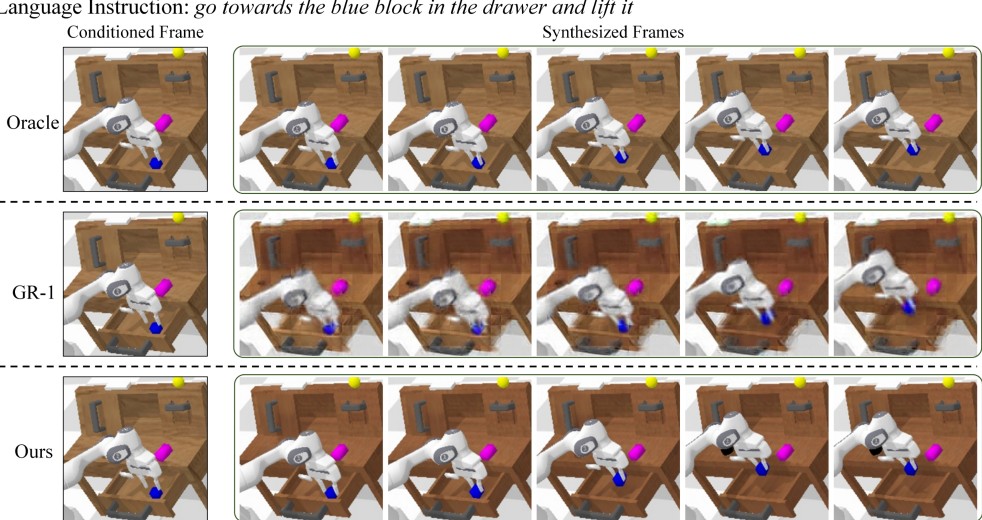

Figure 10: Comparison of goal generation. We visually compare Oracle, GR-1, and our proposed algorithm on the "*go towards the blue block in the drawer and lift it*" task in CALVIN Env. D. Compared with the baseline GR-1, the goal video we generate is more realistic and better restores details.

Language Instruction: *grasp the red block lying in the slider*

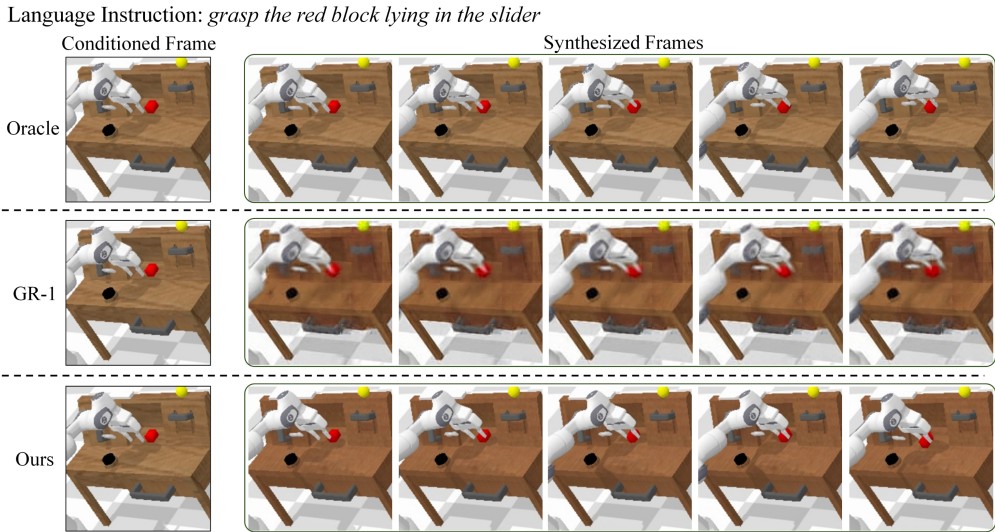

Figure 11: Comparison of goal generation. We visually compare Oracle, GR-1, and our proposed algorithm on the "*grasp the red block lying in the slider*" task in CALVIN Env. D. Compared with the baseline GR-1, the goal video we generate is more realistic and better restores details.

Language Instruction: *go push the red block to the right*

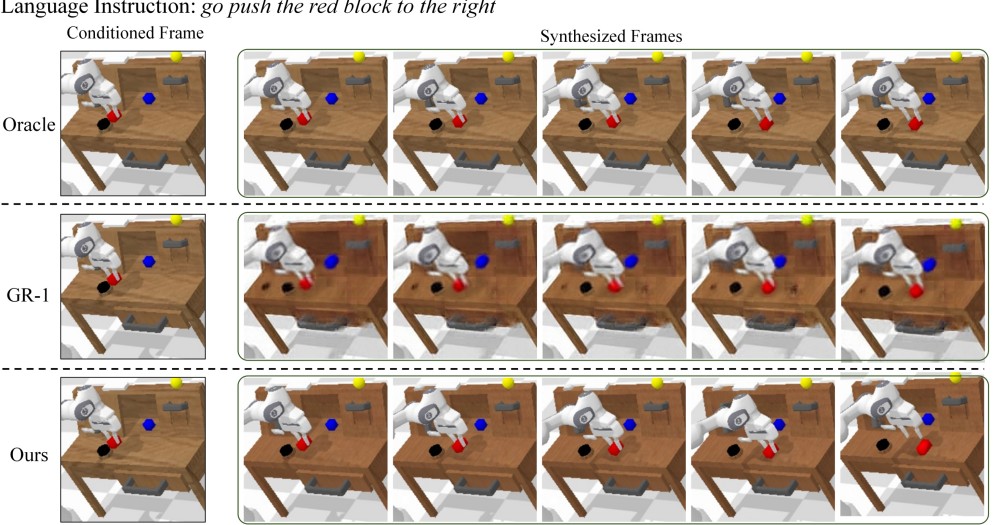

Figure 12: Comparison of goal generation. We visually compare Oracle, GR-1, and our proposed algorithm on the "*go push the red block to the right*" task in CALVIN Env. D. Compared with the baseline GR-1, the goal video we generate is more realistic and better restores details.

Language Instruction: *put pan on stove and put stuffedduck in pan.*

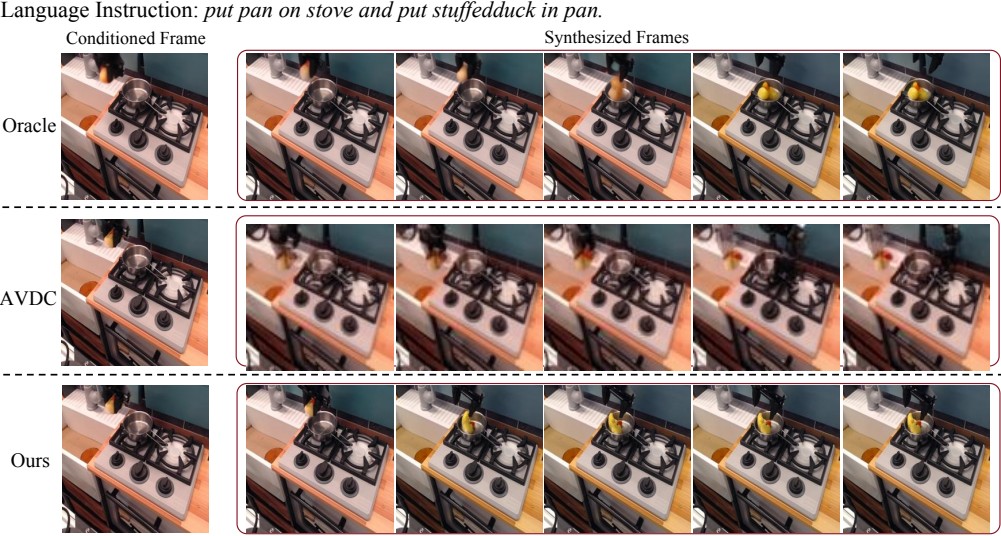

Figure 13: Comparison of goal generation. We visually compare Oracle, AVDC, and our proposed algorithm on the "*put pan on stove and put stuf edduck in pan*" task in Bridge Data. Compared with the baseline AVDC, the goal video we generate is more realistic and better restores details.

Language Instruction: *put yellowpepper on plate and cookiebox in pot or pan on stove*

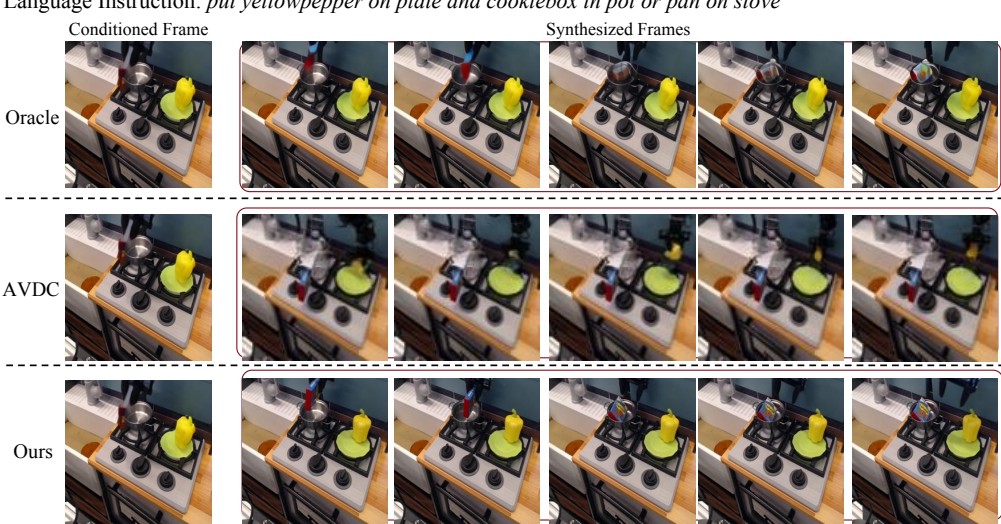

Figure 14: Comparison of goal generation. We visually compare Oracle, AVDC, and our proposed algorithm on the "*put yellowpepper on plate and cookiebox in pot or pan on stove*" task in Bridge Data. Compared with the baseline AVDC, the goal video we generate is more realistic and better restores details.

Language Instruction: *put pot or pan on stove and put egg in pot or pan*

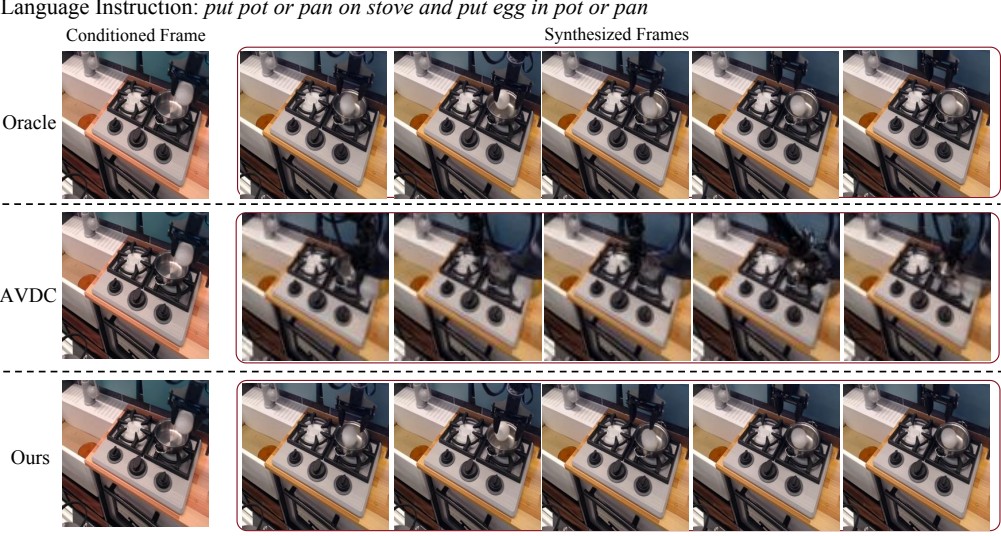

Figure 15: Comparison of goal generation. We visually compare Oracle, AVDC, and our proposed algorithm on the "*put pot or pan on stove and put egg in pot or pan*" task in Bridge Data. Compared with the baseline AVDC, the goal video we generate is more realistic and better restores details.

Table 4: Generalization on perturbed environments in CALVIN (train A, B, C → perturbed test D).

| Perturbed Tasks | Algorithms | No. of Instructions Chained | | | | | Avg. Length (↑) |
|---|---|---|---|---|---|---|---|
| | | 1 | 2 | 3 | 4 | 5 | |
| Image Shift | SuSIE | 0.56 | 0.28 | 0.08 | 0.04 | 0.00 | 0.96 |
| | RoboFlamingo | 0.48 | 0.32 | 0.12 | 0.00 | 0.00 | 0.92 |
| | GR-1 | 0.43 | 0.33 | 0.20 | 0.10 | 0.00 | **1.00** |
| | GEVRM (Ours) | 0.52 | 0.40 | 0.08 | 0.00 | 0.00 | **1.00** |
| Image Rotation | SuSIE | 0.48 | 0.16 | 0.08 | 0.00 | 0.00 | 0.72 |
| | RoboFlamingo | 0.42 | 0.24 | 0.11 | 0.02 | 0.02 | 0.82 |
| | GR-1 | 0.46 | 0.32 | 0.14 | 0.10 | 0.03 | 1.07 |
| | GEVRM (Ours) | 0.60 | 0.32 | 0.12 | 0.08 | 0.04 | **1.16** |
| Color Jitter | SuSIE | 0.72 | 0.36 | 0.16 | 0.12 | 0.08 | 1.44 |
| | RoboFlamingo | 0.52 | 0.22 | 0.08 | 0.08 | 0.04 | 0.94 |
| | GR-1 | 0.6 | 0.35 | 0.21 | 0.12 | 0.07 | 1.35 |
| | GEVRM (Ours) | 0.64 | 0.48 | 0.32 | 0.12 | 0.08 | **1.64** |
| Image Occlusions | SuSIE | 0.72 | 0.48 | 0.32 | 0.32 | 0.24 | 2.08 |
| | RoboFlamingo | 0.43 | 0.30 | 0.13 | 0.06 | 0.03 | 0.96 |
| | GR-1 | 0.78 | 0.60 | 0.46 | 0.32 | 0.23 | 2.39 |
| | GEVRM (Ours) | 0.92 | 0.68 | 0.48 | 0.24 | 0.20 | **2.52** |
| Noise Interference | SuSIE | 0.32 | 0.04 | 0.00 | 0.00 | 0.00 | 0.36 |
| | RoboFlamingo | 0.49 | 0.23 | 0.03 | 0.01 | 0.01 | 0.80 |
| | GR-1 | 0.67 | 0.42 | 0.26 | 0.14 | 0.08 | 1.57 |
| | GEVRM (Ours) | 0.80 | 0.48 | 0.32 | 0.12 | 0.04 | **1.76** |
| Average | SuSIE | 0.56 | 0.26 | 0.13 | 0.10 | 0.06 | 1.11 |
| | RoboFlamingo | 0.63 | 0.35 | 0.18 | 0.09 | 0.05 | 1.31 |
| | GR-1 | 0.67 | 0.38 | 0.22 | 0.11 | 0.06 | 1.44 |
| | GEVRM (Ours) | 0.70 | 0.47 | 0.26 | 0.11 | 0.07 | **1.62** |

Table 5: Zero-shot Generalization. The experiment is set up to train on data from environments A, B, and C (Fig. 4 (**a**)), and test in D (Fig. 4 (**b**)). *: reproduced version on static camera. Static camera: camera of fixed third-person view. Our proposed method can chain more instructions together with a higher success rate than all previous baseline methods. Baseline results are from previous work (Black et al., 2023). The best results for each task are bolded.

| Algorithms | Source | No. of Instructions Chained | | | | |
|---|---|---|---|---|---|---|
| | | 1 | 2 | 3 | 4 | 5 |
| HULC (Mees et al., 2022a) | static camera | 0.43 | 0.14 | 0.04 | 0.01 | 0.00 |
| MCIL (Lynch & Sermanet, 2020) | static camera | 0.20 | 0.00 | 0.00 | 0.00 | 0.00 |
| MdetrLC (Kamath et al., 2021) | static camera | 0.69 | 0.38 | 0.20 | 0.07 | 0.04 |
| AugLC (Pashevich et al., 2019) | static camera | 0.69 | 0.43 | 0.22 | 0.09 | 0.05 |
| LCBC (Walke et al., 2023) | static camera | 0.67 | 0.31 | 0.17 | 0.10 | 0.06 |
| HiP (Ajay et al., 2024) | static camera | 0.08 | 0.04 | 0.00 | 0.00 | 0.00 |
| UniPi (Du et al., 2024) | static camera | 0.56 | 0.16 | 0.08 | 0.08 | 0.04 |
| SuSIE (Black et al., 2023) | static camera | 0.87 | 0.69 | 0.49 | 0.38 | 0.26 |
| GR-1* (Black et al., 2023) | static camera | 0.75 | 0.45 | 0.2 | 0.15 | 0.1 |
| GEVRM (Ours) | static camera | **0.92** | **0.70** | **0.54** | **0.41** | **0.26** |

| Sampling steps | Infer. time [s] | 1 | 2 | 3 | 4 | 5 | Avg. Length |
|---|---|---|---|---|---|---|---|
| 50 | 0.598 | 0.80 | 0.48 | 0.32 | 0.12 | 0.04 | 1.76 |
| 40 | 0.501 | 0.73 | 0.53 | 0.20 | 0.13 | 0.06 | 1.67 |
| 30 | 0.379 | 0.73 | 0.40 | 0.23 | 0.20 | 0.06 | 1.63 |
| 20 | 0.260 | 0.71 | 0.46 | 0.22 | 0.11 | 0.08 | 1.60 |
| 10 | 0.135 | 0.77 | 0.47 | 0.17 | 0.15 | 0.10 | 1.67 |

Table 6: The comparative analysis of the computational efficiency and task success rate of the behavior planner (Noise Interference task). Due to the good properties of the adopted Rectified Flow, when the video sampling steps are reduced, the model inference time is greatly reduced, while the success rate is not significantly reduced.

| Algo. | Control steps | Infer. time [s] | 1 | 2 | 3 | 4 | 5 | Avg. Length |
|---|---|---|---|---|---|---|---|---|
| **DP** | 1 | 0.077 | 0.80 | 0.48 | 0.32 | 0.12 | 0.04 | 1.76 |
| | 2 | 0.044 | 0.85 | 0.50 | 0.20 | 0.15 | 0.05 | 1.75 |
| | 3 | 0.027 | 0.82 | 0.50 | 0.22 | 0.10 | 0.07 | 1.72 |
| | 4 | 0.020 | 0.68 | 0.48 | 0.24 | 0.16 | 0.08 | 1.64 |
| **MLP** | - | 0.019 | 0.73 | 0.40 | 0.13 | 0.06 | 0.06 | 1.40 |

Table 7: Comparison of goal-guided diffusion policies (DP) with different open-loop control steps (Noise Interference task). The results show that the state-aligned policy has better action robustness, and increasing the number of open-loop control steps can significantly reduce the inference time while having little effect on the task success rate. Therefore, the control frequency of our goal-guided diffusion policy can be maintained at the order of tens of Hz, which is sufficient for most robot manipulation tasks in reality. Moreover, when the number of open-loop control steps is 4, the diffusion policy has higher performance, while the inference speed is very close to that of MLP.

Table 8: Behavior planner training hyperparameters.

| Component | Parameter | Value |
|---|---|---|
| Dataset settings | num_frame_total | 51 |
| | transform_name | resize_crop |
| | image_size | (256, 256) |
| Acceleration settings | num_workers | 8 |
| | num_bucket_build_workers | 16 |
| | dtype | bf16 |
| | plugin | zero2 |
| DIT Model settings | type | STDiT3-XL/2 |
| | qk_norm | True |
| | enable_flash_attn | True |
| | enable_layernorm_kernel | True |
| | freeze_y_embedder | True |
| VAE settings | micro_frame_size | 17 |
| | micro_batch_size | 4 |
| Text encoder settings | type | T5 |
| | model_max_length | 300 |
| | shardformer | True |
| Scheduler settings | type | rflow |
| | use_timestep_transform | True |
| | sample_method | logit-normal |
| Random Mask settings | random | 0.025 |
| | intepolate | 0.025 |
| | quarter_random | 0.025 |
| | quarter_head | 0.75 |
| | quarter_tail | 0.025 |
| | quarter_head_tail | 0.05 |
| | image_random | 0.0 |
| | image_head | 0.025 |
| | image_tail | 0.025 |
| | image_head_tail | 0.05 |
| Optimization settings | batch_size | 6 |
| | grad_clip | 1.0 |
| | learning_rate | 1e-4 |
| | ema_decay | 0.99 |
| | adam_eps | 1e-15 |
| | warmup_steps | 1000 |

Table 9: GEVRM test hyperparameters.

| Component | Parameter | Value |
|---|---|---|
| | fixed_interval_number | 20 |
| | condition_frame_length | 5 |
| | goal_generation_number | 51 |
| | micro_frame_size | 17 |
| | num_sampling_steps | 50 |
| General settings | resolution | 256 |
| | aspect_ratio | 1:1 |
| | image_size | (256, 256) |
| | fps | 30 for CALVIN; 5 for Bridge |
| | frame_interval | 1 |
| | dtype | bf16 |

Table 10: Goal-guided policy training hyperparameters.

| Component | Parameter | Value |
|---|---|---|
| | beta_schedule | Cosine |
| | diffusion_steps | 20 |
| | action_samples | 1 |
| General settings | repeat_last_step | 0 |
| | learning_rate | 3e-4 |
| | warmup_steps | 2000 |
| | actor_decay_steps | 2e6 |
| | time_dim | 32 |
| | num_blocks | 3 |
| Score network settings | dropout_rate | 0.1 |
| | hidden_dim | 256 |
| | use_layer_norm | True |
| | hidden_dim | 512 |
| SA Encoder settings | num_prototype | 3000 |
| | temperature | 0.1 |

