# OpenReview forum: "GEVRM: Goal-Expressive Video Generation Model For Robust Visual Manipulation"
_ICLR.cc/2025/Conference — ICLR 2025 Poster_

### Official Review · Reviewer_Ehog · 2024-11-03

**Soundness:** 3
**Presentation:** 3
**Contribution:** 3
**Rating:** 6
**Confidence:** 3

**Summary:**

This paper introduces GEVRM, a novel close-loop VLA model that adopts the Internal Model Control principle to a learning-based framework and improves robustness to perturbations from external environments. This method leverages text-conditioned video generation to infer image goal states, aligns the current and goal states through prototypical contrast learning, and performs goal-guided action prediction using a diffusion policy. Evaluation results demonstrate the strong generalization capability of image goal generation across simulated and real-world datasets, as well as the robustness through high task success rates under external perturbations in the visual manipulation task.

**Strengths:**

1. This paper is well-motivated, clearly written, and easy to follow.
2. The paper properly discussed the related work that leverages text-conditioned video models to generate visual plans for decision-making.
3. GEVRM innovatively uses several techniques to improve the quality of generated visual goals and aligns state representations to enhance the robustness under perturbations
4. The authors provide extensive evaluation results on both behavior planning quality and task completion in perturbed environments, showing the effectiveness of the proposed method.

**Weaknesses:**

1. The underlying framework – composing a text-conditioned video model and an action predictor has been widely explored by previous work, in which most works adopt a relatively lightweight method to extract inverse dynamics. This work employs a diffusion policy for action prediction and interacts with the environment in a close-loop manner, which may help learn multi-modal action distribution but can be time-consuming and often not desirable in real-time deployment.
2. The thoroughness of behavior planner evaluation can be further improved, for example, the effectiveness of techniques like random masking compared to simply masking out the last few frames for decision-making scenarios, should be supported by ablations.
3. It would be great if the authors could thoroughly discuss the limitations of the proposed method.

**Questions:**

1. When GEVRM uses prototypical contrastive learning for state alignment, the positive pairs become the current and goal states from the same trajectory, instead of different augmented views from the same image. I wonder if this contrastive learning process will learn to focus on only trivial consistent features in the background, which are likely to be similar across all timesteps, and ignore some differences (e.g. the position change of objects or the robot arm) between the current and goal states that are essential to help the model make correct decisions.
2. Would the input states be perturbed by any transformations while training the state alignment encoders? If not, are these encoders directly used to process the perturbed images during inference?
3. During the testing time in CALVIN evaluation, apart from the visual discrepancy of the environments, are the instructions for each (sub)task seen during training, or are novel instructions tested to demonstrate task generalization capabilities?

---

> ### Author Response · Authors · 2024-11-25
> **Response to Reviewer Ehog**
>
> **Q1**: This work employs a diffusion policy for action prediction and interacts with the environment in a close-loop manner, which may help learn multi-modal action distribution but can be time-consuming and often not desirable in real-time deployment.
>
> **A1**: To effectively alleviate the problem of low computational efficiency of diffusion models, we propose a goal-guided diffusion policy with multi-step action prediction (4 steps) and allow open-loop control. This setting enables the model to effectively improve computational efficiency while having little effect on the task success rate. For more analysis and experiments, see General Response 3.
>
> **Q2**: The effectiveness of techniques like random masking compared to simply masking out the last few frames for decision-making scenarios, should be supported by ablations.
>
> **A2**: We conducted comparative experiments on the random mask and the mask-only last few frames method, as shown in the following table. The results show that the random mask mechanism significantly improves the overall algorithm.
>
> | **Ablation Study**                | **1**  | **2**  | **3**  | **4**  | **5**  | **Avg. Length** |
> |-----------------------------------|--------|--------|--------|--------|--------|:-----------------:|
> | **Ours (Random Mask)**                | **0.92**   | **0.70**   | **0.54**   | **0.41**   | **0.26**   | **2.83**            |
> | **Masking out the last few frames**   | 0.73   | 0.44   | 0.26   | 0.18   | 0.10   | 1.73            |
>
> **Q3**: It would be great if the authors could thoroughly discuss the limitations of the proposed method.
>
> **A3**: Robust language-based visuomotor control is an important but underexplored field. Our work is just a small step in this direction, and there are still many limitations. In the control tasks of long-term external disturbances, the success rate of the VLA model we proposed still has a lot of room for improvement. How to further deepen the VLA model's understanding of the laws of the physical world and infer stable and reliable (explicit or implicit) behavioral goals is a promising research direction. Moreover, how to integrate these physical knowledge and behavioral goals into the real-time control of the robot in a more efficient and low-cost solution to achieve robust and safe autonomous decision-making is also an important research topic.
>
> Moreover, in the GEVRM model we proposed now, the robot behavior planner and the goal guidance policy are trained separately, which means that it is difficult for the behavior planner model to know the function of the goal guidance policy. Intuitively, by letting the behavior planner know the function of the goal guidance policy, the robustness of the model can be further improved. Therefore, solving this limitation is also an important future work.
>
> **Q4**: I wonder if this contrastive learning process will learn to focus on only trivial consistent features in the background, which are likely to be similar across all timesteps, and ignore some differences (e.g. the position change of objects or the robot arm) between the current and goal states that are essential to help the model make correct decisions.
>
> **A4**: Previous work [1] has demonstrated that contrastive learning can be used to effectively obtain representations of the current and future states of a robot trajectory, so that the representation of the future state is closer to that of random states (derived from other trajectories).
>
> Moreover, in our prototype contrastive learning, we use the Sinkhorn-Knoppal algorithm, which is a fast iterative solution to the entropy-regularized optimal transport problem that considers the Wasserstein distance between high-dimensional data distributions. Compared with the commonly used KL divergence and JS divergence, the Wasserstein distance takes into account the structural information within the probability distribution of the data and can well characterize the distance between distributions even when the "overlap area" is small.
>
> Therefore, some differences between the current state and the target state, such as the position change of an object or a robotic arm, can also be well described by the Wasserstein distance. The visual analysis in Figure 6 of the manuscript shows that after using this method, the state representations between different tasks have better cluster centers and classification boundaries, which can effectively distinguish them.
>
> [1] Eysenbach, B., Zhang, T., Levine, S., & Salakhutdinov, R. R. (2022). Contrastive learning as goal-conditioned reinforcement learning. Advances in Neural Information Processing Systems, 35, 35603-35620.

---

> > ### Author Response · Authors · 2024-11-25
> > **Response to Reviewer Ehog (2)**
> >
> > **Q5**: Would the input states be perturbed by any transformations while training the state alignment encoders? If not, are these encoders directly used to process the perturbed images during inference?
> >
> > **A5**: When training the state alignment encoder, the input state will be disturbed by other transformations. The trained encoder is directly used to handle the disturbed image state during inference without any other operations. This is because the state encoder has already been state aligned, bringing the ability to resist interference into the state representation.
> >
> > **Q6**: During the testing time in CALVIN evaluation, apart from the visual discrepancy of the environments, are the instructions for each (sub)task seen during training, or are novel instructions tested to demonstrate task generalization capabilities?
> >
> > **A6**: To test the generalization ability of the model on new language instructions, we enrich the language setting by generating 50 synonymous instructions for each task (CALVIN ABC-D) using GPT 4. We randomly sample instructions during evaluation. Experimental results show that for generalization to new instructions, our method shows superior performance (average successful task length of 1.99) compared to SuSIE (average successful task length of 1.87).

---

> ### Comment · Reviewer_Ehog · 2024-11-26
> **Rebuttal Acknowledgement**
>
> I appreciate the author's efforts in addressing my concerns and providing additional results. I believe the discussion of inference efficiency and task generalization evaluation will further improve the thoroughness of this work. It would be great if the authors could incorporate these into the final version. I will maintain my positive score.

---

> > ### Author Response · Authors · 2024-11-27
> > **Glad your concern has been addressed**
> >
> > I would like to express my sincere gratitude for your thorough review and constructive feedback on our manuscript. Your insights are very valuable in improving the quality and thoroughness of our work. The relevant experimental results and discussion analysis during the rebuttal period will be reasonably incorporated into the final version of the manuscript in terms of the model's inference efficiency and task generalization evaluation.
> >
> > Thank you again for your continued support and guidance!

---

### Official Review · Reviewer_9iWM · 2024-11-04

**Soundness:** 2
**Presentation:** 3
**Contribution:** 2
**Rating:** 6
**Confidence:** 3

**Summary:**

This paper considers the problem of vision-language-action model for robotics. The proposed approach leverages a text-video generation model to generate the goal images conditioned the current observation and task description. Then the action policy leverages the generated goal and the current state to predict the robot action. To aggress the problem of disturbance encountered during deployment, the paper leverages the idea of IMC from classic control by using internal embeddings (optimized through prototype contrastive learning) as the policy model input.

**Strengths:**

The paper considers an important topic in VLM for robotics. The paper is well-written, and the logic is clear. The proposed approach outperforms the baseline in terms of goal generation when the input image is perturbated and in terms of closed-loop eval in simulations.

**Weaknesses:**

The connection to IMC. The IMC in control uses “diff” between the predicted output from the process model and the actual output from the actual robot, induced by the same action, to generate the feedback signal to compensate the controller. But the proposed approach seems to use the current robot state and the desired output to generate the control signal. I think the proposed structure is just a goal conditioned policy, and I don’t see the strong connection to the IMC.


Image encoder. The image encoder is trained through the discriminating if a goal image and the current state image are from the same trajectory. The experiment is missing ablation studies on the encoder. Would be interested in how an off-the shelf encoder like DinoV2 performs.

**Questions:**

Please see my questions above.

---

> ### Author Response · Authors · 2024-11-25
> **Response to Reviewer 9iWM**
>
> **Q1**: The connection to IMC. The proposed approach seems to use the current robot state and the desired output to generate the control signal. I think the proposed structure is just a goal conditioned policy, and I don’t see the strong connection to the IMC.
>
> **A1**: In classical control theory, the IMC framework is a widely recognized control strategy that uses the internal model of the system to predict future behavior and adjust the control action accordingly, making it highly resistant to interference. The insight of our work is how to effectively implement the idea of IMC principles in the modern learning-based VLA framework to achieve robust control. Our work mainly considers how to flexibly and organically leverage powerful deep learning components to realize the learning-based IMC control principle, rather than using it in a rigid manner.
>
> Therefore, we mainly draw on this broad control idea to design specific learning-based components to resist disturbances during deployment. The robot behavior planner proposed in our work is implemented by a diffusion video generator. By performing prototype contrastive learning on the goal image state and the current image state, the state alignment is achieved and the robust action prediction of the control policy is guided. For more discussion, see General Response 1.
>
> **Q2**: Image encoder. The experiment is missing ablation studies on the encoder. Would be interested in how an off-the shelf encoder like DinoV2 performs.
>
> **A2**: The state encoder we use is of the ResNet type, which has been widely used in previous related work (DP, SuSIE, etc.). Recent work (OpenVLA, etc.) uses off-the-shelf DinoV2 to obtain image features. Our core idea is to enhance the state encoder with prototype contrastive learning, rather than focusing on which encoder to use. Moreover, due to current GPU resources and rebuttal time constraints, a comparative experiment between the two encoders under our model is currently underway. We will update the reply as soon as the latest comparison results are available.

---

> > ### Author Response · Authors · 2024-11-27
> > **Supplement to the answer to question 2**
> >
> > We utilized the off-the-shelf DINOv2 as the image encoder and train a goal-guided policy. Experiments show that the average success sequence length on the perturbed CALVIN ABC-D task is zero. We find that during the training phase, the training loss of the DINOv2-based policy can only converge to about 0.3, while the training loss of the ResNet-based policy can converge to about 0.03. Moreover, the concurrent work [1] shows in the section “6 MODEL SIZE AND TRAINING STRATEGY: BEYOND DATA SCALING” that “... As shown in Table 2a, a Learning-from-Scratch (LfS) ViT-L/14 and the use of frozen DINOv2 pretrained features achieve scores close to zero.”. Therefore, their experimental results (Table 2 (a)) also show that using the off-the-shelf DINOv2 as the image encoder will also lead to poor performance.
> >
> > We speculate that the DINOv2 pretrained on the extensive Internet image data contains general semantic features, which have a gap with the robot trajectory state features. This makes it difficult to obtain accurate action outputs through a simple policy network. Instead, it needs to be processed through a more expressive network in OpenVLA to obtain the final effective action.
> >
> > [1] Lin, F., Hu, Y., Sheng, P., Wen, C., You, J., & Gao, Y. (2024). Data Scaling Laws in Imitation Learning for Robotic Manipulation. arXiv preprint arXiv:2410.18647.

---

> ### Author Response · Authors · 2024-11-30
> **Sincerely hope for further feedback**
>
> Dear reviewer 9iWM:
>
> For your main concerns, such as connection with traditional IMC, comparison of image encoders, etc., we have explored the relationship with traditional IMC in depth and conducted relevant comparative experiments. We are sure that **we have addressed all your questions and concerns**.
>
> The interactive discussion period is about to end, but we have not received further feedback. We hope that we can have more discussions and exchanges with you in time and convince you to consider **raising the score from negative to positive**. Please feel free to ask for any additional information or clarification that may be needed.
>
> Thank you for taking the time to give insightful feedback in your busy schedule.

---

> ### Author Response · Authors · 2024-12-02
> **Desire for further feedback**
>
> Dear Reviewer 9iWM:
>
> The deadline for the rebuttal is approaching. We are eager to have you spare a little time from your busy schedule to provide valuable feedback. We are sure that **we have addressed all your questions and concerns**. We sincerely hope to get your further feedback and convince you to consider **raising the rating from negative to positive**.

---

> > ### Comment · Reviewer_9iWM · 2024-12-02
> >
> > I would like to thank the reviewer for their feedback! I still think the proposed is a bit unnecessary to connect to IMC. In light of the additional strong results. I will update the rating.

---

> > > ### Author Response · Authors · 2024-12-02
> > > **Glad to receive your prompt feedback and score improvement**
> > >
> > > I would like to express my sincere gratitude for your insightful and constructive feedback on our work. It is great to see that the strong results have contributed to your score improvement. Regarding the connection of our work to IMC, our insight lies in borrowing classical IMC ideas in the framework of modern learning-based VLAs to achieve robust actions and resist perturbations. In the specific implementation process, we need to consider the functions of various learning-based components in a flexible and effective way rather than a strict and rigid way.
> > >
> > > Thank you again for your positive feedback and valuable comments.

---

### Official Review · Reviewer_siH9 · 2024-11-04

**Soundness:** 3
**Presentation:** 3
**Contribution:** 2
**Rating:** 6
**Confidence:** 5

**Summary:**

This paper presents a goal-guided policy conditioned on future predictions generated by a video diffusion model. The authors try to integrate the classic principle of internal model control (IMC) into visuomotor control, thereby enhancing the robot's capability to resist environmental perturbations.

**Strengths:**

1. The method appears to be reasonably designed. It utilizes a video diffusion model as a visual planner to generate future predictions that guide the policy in action output. In terms of detailed design, the proposed method enhances the generation consistency through efficient video spatiotemporal compression and random mask strategies. Furthermore, the authors utilize prototypical contrastive learning to align the goal states with its current state, thereby enabling the model to implicitly infer and distinguish perturbations from the external environment.
2. The authors propose generalization experiments on perturbed environments in CALVIN (train A, B, C → perturbed test D), which can assist the research community in examining the the robustness of policies against external environmental disturbances.

**Weaknesses:**

1. This paper did not undertake real-world robotic experiments. Instead, it focused exclusively on analyzing video prediction quality using the Bridge dataset and conducting action execution tests within the simulated environment of the Calvin benchmark. However, the effectiveness of GEVRM on real-world robots was not validated.
2. In Table 2, the authors compare their proposed GEVRM method with other baseline approaches within the CALVIN ABC-D environment, utilizing the settings of RGB input only. However, they did not involve powerful baseline methods for comparison. While GR-1 is a comparatively strong baseline, its reproduced performance is notably low. Could the authors provide further details regarding the reproduction of GR-1? Furthermore, would it be possible for the authors to incorporate additional powerful methods for comparison by adjusting their input modalities to RGB only? Additionally, I would like to mention that Susie's results on Calvin ABC-D should be included in Table 2.
3. The authors should give more credits to the paper "Closed-Loop Visuomotor Control with Generative Expectation for Robotic Manipulation" (https://arxiv.org/abs/2409.09016). The design philosophy behind GEVRM and the presentation of the article, such as the layout in Figure 1, seem to be inspired by this paper; however, it is not mentioned in the introduction or related works sections.

**Questions:**

1. The integration of a video diffusion model might introduce considerable computational overhead. However, the paper lacks a comprehensive analysis of the computational requirements. How about the inference efficiency in Calvin simulation environment?
2. On the goal-conditioned policy, the authors propose a diffusion policy conditioned on the generated goals. In such goal-conditioned policy framework, does the diffusion policy exhibit significant superiority compared to a simple MLP?
3. In Table 3, the generalization experiments are conducted in perturbed environments. How do the other baseline methods, apart from Susie, perform in this setting?

---

> ### Author Response · Authors · 2024-11-25
> **Response to Reviewer siH9**
>
> **Q1**: The effectiveness of GEVRM on real-world robots was not validated.
>
> **A1**: To verify the effectiveness of the proposed GEVRM in real-world robot manipulation tasks, we collected more than 400 expert trajectories for picking and placing cups, bowls, and tiger plush toys on a real robotic arm UR5. We trained GEVRM based on these real data and tested it in real scenarios. The results show that GEVRM can be effectively deployed to pick and place tasks of common objects in real scenarios. The specific real-world experiments of GEVRM have been supplemented in the appendix section of the manuscript “Real-World Tasks.”:
>
> “Protocol. To examine the effectiveness of the proposed GEVRM on real-world robotic manipulation tasks, we propose a real-machine deployment protocol. We evaluate GEVRM on a robotic arm UR5 for the pick-and-place tasks of a cup, a bowl, and a tiger plush toy. Specifically, we use a camera to capture third-person images as the observation space (image width 640, height 480), and relative poses and binarized gripper states as the action space (7 dimensions). The total number of collected real-world teleoperation expert trajectories is over 400, with trajectory lengths ranging from 20 to 120 steps and a control frequency of 5Hz.
>
> Experiments. We train and evaluate GEVRM under real-world protocols. The VAE and DiT in the behavior planner are trained for 30,000 and 12,000 iterations, respectively, while the goal-guided policy is trained for 100,000 iterations. Other hyperparameters remain the same as in the experiments in CALVIN (and Bridge). Fig. 7 shows the policy execution process of our proposed GEVRM on three types of real-world tasks, indicating that our method can be effectively deployed on real machines. In terms of task success rate (SR), we evaluated each type of task 10 times. The experimental results show that compared with the grasping and placing of cups (or bowls) with regular shapes (success rate of about 0.8), the grasping of tiger plush toys with soft materials and irregular shapes is more challenging (success rate of about 0.6). Further improving GEVRM's perception of real-world scenes and task execution accuracy is an important future work.”
>
> **Q2**: Could the authors provide further details regarding the reproduction of GR-1? Furthermore, would it be possible for the authors to incorporate additional powerful methods for comparison by adjusting their input modalities to RGB only? Additionally, I would like to mention that Susie's results on Calvin ABC-D should be included in Table 2.
>
> **A2**: **GR-1 reproduction details:** For the sake of fair experimental comparison, we only use third-person images, without first-person and proprioceptive information as model input. At the same time, when predicting future image frames, only third-person images are considered, that is, future images from the first person are not predicted. Other pre-trained models and hyperparameter configurations are consistent with the original GR-1 paper.
>
> **More baseline comparisons:** The main problem studied in our work is how to make VLA produce robust actions in a perturbed environment, so the core lies in goal state planning and action execution in a perturbed environment. For the goal state planning task (Table 1 of the manuscript), we added SuSIE. The SuSIE results have been added to Table 2 of the revised manuscript. For the perturbed CALVIN ABC-D environment generalization task (Table 3 of the manuscript), we added RoboFlamingo and GR-1. Experiments show that our method achieves better performance, see General Response 2.
>
> **Q3**: The authors should give more credits to the paper "Closed-Loop Visuomotor Control with Generative Expectation for Robotic Manipulation" … however, it is not mentioned in the introduction or related works sections.
>
> **A3**: We added a description in the “Internal Model Control (IMC) framework” section of the Related Work section of the manuscript, summarizing the main research content of this work: “… More recently, inspired by classical closed-loop control systems, a closed-loop visuomotor control framework has been proposed that incorporates feedback mechanisms to improve adaptive robot control.”
>
> **Q4**: The paper lacks a comprehensive analysis of the computational requirements. How about the inference efficiency in Calvin simulation environment?
>
> **A4**: Our work proposes corresponding strategies from three aspects to effectively alleviate the problem of low computational efficiency of diffusion models:
>
> (1) Use Rectified Flow instead of DDPM to train the behavior planner;
>
> (2) During inference, call the behavior planner at a fixed step interval (20 steps);
>
> (3) Goal-guided diffusion multi-step action prediction (4 steps) allows open-loop control.
>
> All these settings enable the model to effectively improve the computational efficiency without having much effect on the task success rate. For detailed analysis and experiments, see General Response 3.

---

> > ### Author Response · Authors · 2024-11-25
> > **Response to Reviewer siH9 （2）**
> >
> > **Q5**: In such goal-conditioned policy framework, does the diffusion policy exhibit significant superiority compared to a simple MLP?
> >
> > **A5**: As shown in the table below, we compare the performance of goal-guided diffusion policy (Ours) and MLP on all perturbation tasks. The results show that our method outperforms MLP on all perturbation tasks.
> >
> > | **Perturbed Tasks**  | **Algorithms** | **Avg. Length** |
> > |----------------------|:----------------:|:-----------------:|
> > | **Image Shift**          | MLP            | 0.80            |
> > |                      | Ours           | **1.00**            |
> > | **Image Rotation**       | MLP            | 1.00            |
> > |                      | Ours           | **1.16**            |
> > | **Color Jitter**         | MLP            | 1.56            |
> > |                      | Ours           | **1.64**            |
> > | **Image Occlusions**     | MLP            | 2.26            |
> > |                      | Ours           | **2.52**            |
> > | **Noise Interference**   | MLP            | 1.40            |
> > |                      | Ours           | **1.76**            |
> > | **Average**              | MLP            | 1.40            |
> > |                      | Ours           | **1.62**            |
> >
> > **Q6**: In Table 3, the generalization experiments are conducted in perturbed environments. How do the other baseline methods, apart from Susie, perform in this setting?
> >
> > **A6**: We add RoboFlamingo (pure imitation learning) and GR-1 algorithms using data augmentation in the perturbed CALVIN ABC-D generalization task. The experimental results show that our proposed method GEVRM surpasses the previous baseline algorithms. See General response 2 for more experimental details.

---

> ### Author Response · Authors · 2024-11-30
> **Sincerely hope for further feedback**
>
> Dear reviewer siH9:
>
> For your main concerns, such as the lack of real machine experiments and high computational overhead, we have supplemented real machine experiments, and efficiency comparison experiments, and improved the manuscript.  We are confident that **all your questions and concerns have been addressed**.
>
> The interactive discussion period is about to end, but we have not received further feedback. We hope that we can have more discussions and exchanges with you in time and convince you to consider **raising the score from negative to positive**. Please feel free to ask for any additional information or clarification that may be needed.
>
> Thank you for taking the time to provide insightful feedback.

---

> > ### Comment · Reviewer_siH9 · 2024-11-30
> >
> > Thank you for the considerable effort you put into addressing my concerns, particularly through the supplemented real-world robotic experiments and further comparative studies, which have resolved most of my issues. I have some additional questions about the real-world experiments supplemented during the rebuttal phase.
> >
> > 1. What is the paradigm of the 10-time evaluation, and does it involve position generalization?
> >
> > 2. How well do the real-world policy perform in maintaining stability under perturbations like those in figure 4?

---

> > > ### Author Response · Authors · 2024-11-30
> > > **Responses to two additional questions**
> > >
> > > We are happy to receive your feedback and see that most of your questions have been addressed. For your additional two questions about real-world experiments, our responses are as follows:
> > >
> > > **Q1**: What is the paradigm of the 10-time evaluation, and does it involve position generalization?
> > >
> > > **A1**: In our real-world tasks, there are three types of manipulation tasks according to the types of objects (cups, bowls, and plush toys). Each type of task is repeated 10 times, and the positions of each object are manually randomly adjusted at the beginning of each task. When the robot executes the action inferred by the model for a given time (60 seconds for cup and bowl tasks, 90 seconds for plush toy tasks), we judge whether the task is successful. Finally, we count the number of successes and calculate the success rate. This evaluation involves position generalization.
> > >
> > > **Q2**: How well do the real-world policy perform in maintaining stability under perturbations like those in figure 4?
> > >
> > > **A2**: In real-world experiments, we tested the performance of the model when the camera view was rotated within a certain range (about 25 degrees). The results show that the number of successes only decreases by one or two. Therefore, the model still has good stability under perturbations and its performance does not drop significantly.
> > >
> > > Thank you for your question about real-world experiments. Please let us know if there are any other concerns that prevent your score from improving from negative to positive.

---

> > > > ### Comment · Reviewer_siH9 · 2024-12-02
> > > >
> > > > The supplementary experiments and responses presented during the rebuttal phase have further validated the effectiveness of GEVRM. I decide to raise my score.

---

> > > > > ### Author Response · Authors · 2024-12-02
> > > > > **Glad to see your concerns addressed and the score improved**
> > > > >
> > > > > Sincerely thank you for your positive response. It is great to see that our additional supplementary experiments and the responses during the rebuttal period have completely addressed your questions and concerns. Your thorough review and constructive feedback on our manuscript have greatly improved the quality and completeness of our work.

---

### Official Review · Reviewer_TJEW · 2024-11-04

**Soundness:** 3
**Presentation:** 2
**Contribution:** 3
**Rating:** 6
**Confidence:** 4

**Summary:**

This paper introduces GEVRM, a method for robust visual manipulation through video generation. It trains a video generator for goal generation, and a goal-conditioned behavioral cloning policy. The video generator takes a sequence of observations and a language instruction, and predicts the subsequent image observations as the goal. The policy head is a goal-conditioned diffusion policy. GEVRM uses an auxiliary state alignment loss to improve learned representations. Results show that GEVRM outperforms previous methods on Bridge and CALVIN data, and is more robust to perturbations and generalizable than baselines in the simulated environment CALVIN.

**Strengths:**

- Video generation for goal-conditioned behavioral cloning is well-motivated.
- GEVRM outperforms previous methods on Bridge and CALVIN for video generation.
- GEVRM outperforms HiP, UniPi, and GR-1 on generalization in CALVIN, and outperforms SuSIE on generalization with visual perturbations.

**Weaknesses:**

- Limited/inconsistent evaluation. GEVRM is compared with AVDC on Bridge and with GR-1 on CALVIN for video generation, with HiP/UniPi/GR-1 on unperturbed generalization, and with SuSIE on perturbed generalization. It would be good to see GEVRM compared with a consistent set of baselines for video generation and policy rollouts in a few more environments.
- If the core problem setting is robust manipulation, it would be convincing to see GEVRM compared with HiP, UniPi, GR-1, and pure behavioral cloning, each with data augmentation.
- It is unclear what the key contributions of GEVRM are and why it outperforms prior work. A more comprehensive ablation with downstream task success rate could justify each component in the design of GEVRM and clarify its improvement over prior state of the art.

**Questions:**

- Baseline (video generation): is AVDC/GR-1/GEVRM trained with data augmentation?
- Baseline (CALVIN rollout): how does GEVRM compare to HiP/UniPi/GR-1/behavioral cloning with data augmentation?
- Ablation: how does setting the state alignment loss $\lambda = 0$ affect task success rate? Quantitative results in addition to the t-SNE plot would be convincing.
- Ablation/Clarification: video generation for control and goal-conditioned behavioral cloning are relatively well-studied problems. Could you clarify the key contribution of GEVRM? (How is GEVRM different from prior work in its architecture, and what design choices contributes to the improved performance?) A more comprehensive ablation studying the importance of each component in GEVRM would be convincing to see.

---

> ### Author Response · Authors · 2024-11-25
> **Response to Reviewer TJEW**
>
> **Q1**: It would be good to see GEVRM compared with a consistent set of baselines for video generation and policy rollouts in a few more environments. It would be convincing to see GEVRM compared with HiP, UniPi, GR-1, and pure behavioral cloning, each with data augmentation.
>
> **A1**: We added the SuSIE algorithm to the goal generation task; added the SuSIE algorithm to the CALVIN ABC-D generalization task; added RoboFlamingo (pure imitation learning) and GR-1 algorithms using data augmentation in the perturbed CALVIN ABC-D generalization task. Considering the extremely poor performance of HiP in the CALVIN ABC-D generalization task (the average success length is very close to 0), its implementation in the perturbed environment is not necessary. **In these three types of tasks, our method GEVRM shows superior performance.** For more experimental details, see General Response 2.
>
> **Q2**: Could you clarify the key contribution of GEVRM? How is GEVRM different from prior work in its architecture, and what design choices contributes to the improved performance?
>
> **A2**: The **core contribution** of our work is to propose the necessary components in the VLA framework that can effectively implement the IMC principle (mainly including **goal state generation** and **goal alignment**), thereby improving the robustness of decision actions to resist external interference during deployment.
>
> Comparison with the structure of previous works and design choices: The network structure used in previous works: AVDC (Clip+U-Net+DDIM), HiP (GPT3.5-turbo+PVDM+ViT-B), UniPi (T5-XXL+U-Net), GR-1 (Clip+ViT+GPT-style Transformer+MSE), Roboflamingo (ViT+Flamingo+LSTM+MSE) . Different from these works, when we instantiate IMC in VLA, we consider the structure of **T5-XXL+DiT+2DVAE & 3D Causal VAE + Rectified Flow**. The main designs include using 2D & 3D Causal VAE to achieve efficient compression encoding of robot image sequences, random masking mechanism for effective understanding of object dynamics and time series correlation, and prototype contrastive learning for state alignment (SA) to simulate responses. The quantitative ablation comparison results of each component are shown in the table below. See General Response 1 for more discussion.
> | **Design choices**              | **1**  | **2**  | **3**  | **4**  | **5**  | **Avg. Length** |
> |----------------------------------|--------|--------|--------|--------|--------|:-----------------:|
> | Ours                             | 0.92   | 0.70   | 0.54   | 0.41   | 0.26   | 2.83            |
> | Ours w/o finetune VAE            | 0.81   | 0.59   | 0.40   | 0.31   | 0.21   | 2.32 (-22%)      |
> | Ours w/o SA                      | 0.86   | 0.68   | 0.46   | 0.30   | 0.26   | 2.56 (-10.5%)    |
> | Ours w/o random mask             | 0.73   | 0.44   | 0.26   | 0.18   | 0.10   | 1.73 (-63.6%)    |
>
> **Q3**: Baseline (video generation): is AVDC/GR-1/GEVRM trained with data augmentation?
>
> **A3**: The experimental results in Table 1 of the manuscript are mainly used to verify the ability of various methods (AVDC/GR-1/GEVRM) to generate robot image goal states, so data augmentation methods are not used. Moreover, for experiments on policy execution, these models use data augmentation methods.
>
> **Q4**: How does setting the state alignment loss λ=0 affect task success rate? Quantitative results in addition to the t-SNE plot would be convincing.
>
> **A4**: The ablation experiment of state alignment (SA) loss λ=0 (Ours w/o SA) has been shown in Figure 5 of the manuscript. For a clearer comparison, the relevant numerical results are summarized in the following table. After using SA, the metric Avg. Length increases from 2.56 to 2.83, bringing a 10.5% performance improvement. This proves the effectiveness of the SA component in the proposed algorithm GEVRM.
>
> | **Ablation Study**               | **1**  | **2**  | **3**  | **4**  | **5**  | **Avg. Length** |
> |----------------------------------|--------|--------|--------|--------|--------|-----------------|
> | **Ours w/o SA (λ=0)**                | 0.86   | 0.68   | 0.46   | 0.30   | **0.26**   | 2.56            |
> | **Ours (λ=1)**                       | **0.92**   | **0.70**   | **0.54**   | **0.41**   | **0.26**   | **2.83 (+10.5%)**    |

---

> > ### Comment · Reviewer_TJEW · 2024-11-30
> > **Thanks for the additional experiments and insights**
> >
> > Thanks for the additional experiments and insights. The additional ablation experiments and rebuttal response have clarified the core contribution of GEVRM. I have updated my rating.

---

> > > ### Author Response · Authors · 2024-11-30
> > > **Glad to see your concerns addressed and the score improved**
> > >
> > > Sincerely thank you for your prompt response. Glad to see that our additional ablation experiments and in-depth elaboration of the core contribution fully addressed your questions and concerns. Your thorough review and constructive feedback on our manuscript greatly enhanced the quality and completeness of our work.

---

> ### Author Response · Authors · 2024-11-30
> **Sincerely hope for further feedback**
>
> Dear reviewer TJEW:
>
> For your main concerns, such as inconsistent evaluation and unclear main contributions, we have improved the relevant experiments and contribution descriptions. We are confident that **all your questions and concerns have been addressed**.
>
> The interactive discussion period is about to end, but we have not received further feedback. We hope that we can have more discussions and exchanges with you in time and convince you to consider **raising the score from negative to positive**. Please feel free to ask for any additional information or clarification that may be needed.
>
> Thank you for taking the time to give insightful feedback in your busy schedule.

---

### Author Response · Authors · 2024-11-25
**General Response (1)**

We would like to thank the reviewers for their appreciation of our work. Reviewers TJEW and Ehog recognized our method was "**well-motivated**", and reviewers 9iWM and Ehog appreciated our manuscript was "**clearly written**", "**logically clear**" and "**easy to follow**". We also thank all the reviewers for their professional guidance, constructive comments, and insightful suggestions, which enabled us to improve the manuscript further. We have thoroughly performed additional experiments and revised the manuscript to address all your comments and concerns.
## 1. The core contribution of our method.
Our **motivation** is that VLA models trained in ideal environment data are inevitably subject to external perturbations when deployed, resulting in fragile and unstable actions, which leads to a significant decrease in their generalization performance. We aim to explore how to instantiate the classic internal model control (IMC) principle in the VLA framework to improve the robustness of decision-making actions. Therefore, **the core contribution of our work lies in the necessary components in the VLA framework, mainly including goal state generation and goal alignment**. The widespread principle of IMC suggests that a closed-loop system with an internal model that includes external input signals can accurately track the reference input and effectively cancel out disturbances. We, therefore, borrow this control idea for VLA robust action generation. After the DiT video model generates the goal state, prototype contrastive learning is utilized to enhance the state representation to compensate the low-level controller.
## 2. Comparison with more advanced baseline algorithms.
To compare with the baseline algorithms more comprehensively, we added the SuSIE algorithm to the goal generation task (Table 1 of the manuscript); added the SuSIE algorithm to the CALVIN ABC-D generalization task (Table 2 of the manuscript); added RoboFlamingo (pure imitation learning) and GR-1 algorithms to the perturbed CALVIN ABC-D generalization task (Table 3 of the manuscript). The specific results are shown in the following three tables. The results show that compared with the baseline algorithms SuSIE and GR-1, our method GEVRM shows superior performance in the goal generation task, the environment generalization task, and the perturbed environment generalization task. In addition, SuSIE is a method based on single-frame image state generation, while HiP, UniPi and our GEVRM are methods based on multi-frame video state generation. Therefore, our task is more challenging, aiming to allow the model to learn more temporal consistency of the robot arm and object motion.

### Goal generation task
| **Benchmark**  | **Algorithms** | **FID (↓)**         | **FVD (↓)**         | **LPIPS (↓)**         | **SSIM (↑)**        | **PSNR (↑)**        |
|:--------------:|:--------------:|:-------------------:|:-------------------:|:---------------------:|:-------------------:|:-------------------:|
| **BridgeData**  | AVDC           | 246.45 ± 39.08      | 22.89 ± 4.99        | 0.23 ± 0.03           | 0.73 ± 0.05         | 18.22 ± 2.53        |
| **BridgeData**  | SuSIE          | 114.79 ± 21.38      | -                   | 0.217 ± 0.082         | 0.706 ± 0.070       | 16.388 ± 2.901      |
| **BridgeData**  | GEVRM (Ours)   | **35.70 ± 10.77**   | **4.16 ± 1.35**     | **0.06 ± 0.03**       | **0.89 ± 0.04**     | **22.36 ± 2.75**    |
| **CALVIN**      | GR-1           | 236.75 ± 38.87      | 12.83 ± 2.60        | 0.20 ± 0.02           | 0.65 ± 0.03         | 18.59 ± 0.95        |
| **CALVIN**      | SuSIE          | 214.14 ± 45.45      | -                   | 0.150 ± 0.041         | 0.750 ± 0.045       | 18.115 ± 2.289      |
| **CALVIN**      | GEVRM (Ours)   | **94.47 ± 22.54**   | **3.80 ± 1.20**     | **0.09 ± 0.04**       | **0.80 ± 0.05**     | **21.10 ± 3.29**    |

### CALVIN ABC-D generalization task (Only static camera is used)
| **Algorithms** | **1**   | **2**   | **3**   | **4**   | **5**   |
|-----------------|:---------:|:---------:|:---------:|:---------:|:---------:|
| **HiP**            | 0.08    | 0.04    | 0.00    | 0.00    | 0.00    |
| **UniPi**          | 0.56    | 0.16    | 0.08    | 0.08    | 0.04    |
| **GR-1**           | 0.75    | 0.45    | 0.20    | 0.15    | 0.10    |
| **SuSIE**          | 0.87    | 0.69    | 0.49    | 0.38    | 0.26    |
| **GEVRM (Ours)**   | **0.92**    | **0.70**    | **0.54**    | **0.41**    | **0.26**    |

---

> ### Author Response · Authors · 2024-11-25
> **General Response (2)**
>
> ### Perturbed CALVIN ABC-D generalization task (Only static camera is used)
>
> | **Perturbed Tasks**   | **Algorithms**   | **1**  | **2**  | **3**  | **4**  | **5**  | **Avg. Length** |
> |-----------------------|------------------|:--------:|:--------:|:--------:|:--------:|:--------:|:-----------------:|
> | **Image Shift**       | SuSIE        | 0.56   | 0.28   | 0.08   | 0.04   | 0.00   | 0.96            |
> |                       | RoboFlamingo | 0.48   | 0.32   | 0.12   | 0.00   | 0.00   | 0.92            |
> |                       | GR-1         | 0.43   | 0.33   | 0.20   | 0.10   | 0.00   | **1.00**            |
> |                       | Ours         | 0.52   | 0.40   | 0.08   | 0.00   | 0.00   | **1.00**            |
> | **Image Rotation**     | SuSIE        | 0.48   | 0.16   | 0.08   | 0.00   | 0.00   | 0.72            |
> |                        | RoboFlamingo | 0.42   | 0.24   | 0.11   | 0.02   | 0.02   | 0.82            |
> |                        | GR-1         | 0.46   | 0.32   | 0.14   | 0.10   | 0.03   | 1.07            |
> |                        | Ours         | 0.60   | 0.32   | 0.12   | 0.08   | 0.04   | **1.16**            |
> | **Color Jitter**       | SuSIE        | 0.72   | 0.36   | 0.16   | 0.12   | 0.08   | 1.44            |
> |                       | RoboFlamingo | 0.52   | 0.22   | 0.08   | 0.08   | 0.04   | 0.94            |
> |                       | GR-1         | 0.60   | 0.35   | 0.21   | 0.12   | 0.07   | 1.35            |
> |                       | Ours         | 0.64   | 0.48   | 0.32   | 0.12   | 0.08   | **1.64**            |
> | **Image Occlusions**   | SuSIE        | 0.72   | 0.48   | 0.32   | 0.32   | 0.24   | 2.08            |
> |                       | RoboFlamingo | 0.43   | 0.30   | 0.13   | 0.06   | 0.03   | 0.96            |
> |                       | GR-1         | 0.78   | 0.60   | 0.46   | 0.32   | 0.23   | 2.39            |
> |                       | Ours         | 0.92   | 0.68   | 0.48   | 0.24   | 0.20   | **2.52**            |
> | **Noise Interference** | SuSIE        | 0.32   | 0.04   | 0.00   | 0.00   | 0.00   | 0.36            |
> |                       | RoboFlamingo | 0.49   | 0.23   | 0.03   | 0.01   | 0.01   | 0.80            |
> |                       | GR-1         | 0.67   | 0.42   | 0.26   | 0.14   | 0.08   | 1.57            |
> |                       | Ours         | 0.80   | 0.48   | 0.32   | 0.12   | 0.04   | **1.76**            |
> | **Average**            | SuSIE        | 0.56   | 0.26   | 0.13   | 0.10   | 0.06   | 1.11            |
> |                       | RoboFlamingo | 0.63   | 0.35   | 0.18   | 0.09   | 0.05   | 1.31            |
> |                       | GR-1         | 0.67   | 0.38   | 0.22   | **0.11**   | 0.06   | 1.44            |
> |                       | Ours         | **0.70**   | **0.47**   | **0.26**   | **0.11**   | **0.07**   | **1.62**            |
>
> ## 3. Inference efficiency of the proposed method.
> To effectively alleviate the problem of low computational efficiency of diffusion models, our work proposes corresponding strategies from three aspects:
>
> (1) During the training phase of the behavior planner, we utilize the currently advanced Rectified Flow instead of DDPM to train the video generation model. Rectified Flow promotes the learning of the mapping from noise to the real image distribution by solving ordinary differential equations along the straight path between samples. This method has been proven to be a more effective training paradigm that can significantly reduce the video sampling steps, thereby improving the model training speed and reducing its inference time.
>
> (2) During the test phase, we call the behavior planner only after a fixed interval of the lower-level goal-guided diffusion policy execution step (fixed to 20 for all experiments) instead of every step.
>
> (3) When training the goal-guided diffusion policy, we predict multiple steps (fixed to 4 for all experiments) instead of a single-step action. This allows us to perform open-loop control during the test phase to increase computational efficiency.
>
> The comparative analysis of the computational efficiency and task success rate of the behavior planner is shown in the following table (Noise Interference task). Due to the good properties of the adopted Rectified Flow, when the video sampling steps are reduced, the model inference time is greatly reduced, while the success rate is not significantly reduced.

---

> > ### Author Response · Authors · 2024-11-25
> > **General Response (3)**
> >
> > ### Diffusion video goal generation efficiency
> > | **Sampling steps** | **Inference time** | **1**  | **2**  | **3**  | **4**  | **5**  | **Avg. Length** |
> > |:--------------------:|:--------------:|:--------:|:--------:|:--------:|:--------:|:--------:|:------------:|
> > | **50**          | 0.598              | 0.80   | 0.48   | 0.32   | 0.12   | 0.04   | 1.76            |
> > | **40**          | 0.501              | 0.73   | 0.53   | 0.20   | 0.13   | 0.06   | 1.67            |
> > | **30**          | 0.379              | 0.73   | 0.40   | 0.23   | 0.20   | 0.06   | 1.63            |
> > | **20**          | 0.260              | 0.71   | 0.46   | 0.22   | 0.11   | 0.08   | 1.60            |
> > | **10**          | 0.135              | 0.77   | 0.47   | 0.17   | 0.15   | 0.10   | 1.67            |
> >
> > We also conducted experimental comparisons on goal-guided diffusion policies with different open-loop control steps, as shown in the table below (Noise Interference task). The results show that the state-aligned policy has better action robustness, and increasing the number of open-loop control steps can significantly reduce the inference time while having little effect on the task success rate. Therefore, the control frequency of our goal-guided diffusion policy can be maintained at the order of tens of Hz, which is sufficient for most robot manipulation tasks in real scenarios. Moreover, when the number of open-loop control steps is 4, the diffusion policy has higher performance, while the inference speed is very close to that of MLP.
> > ### Diffusion Policy Inference efficiency
> > |  | **Open-loop control steps** | **Inference time** | **1**  | **2**  | **3**  | **4**  | **5**  | **Avg. Length** |
> > |----------------------|:--------------:|:---------------:|--------|--------|--------|--------|--------|:-----------------:|
> > | **Diffusion policy**       | 1                           | 0.077              | 0.80   | 0.48   | 0.32   | 0.12   | 0.04   | 1.76            |
> > |                      | 2                           | 0.044              | 0.85   | 0.50   | 0.20   | 0.15   | 0.05   | 1.75            |
> > |                      | 3                           | 0.027              | 0.82   | 0.50   | 0.22   | 0.10   | 0.07   | 1.72            |
> > |                      | 4                           | 0.020              | 0.68   | 0.48   | 0.24   | 0.16   | 0.08   | 1.64            |
> > | **MLP**              |                             | 0.019              | 0.73   | 0.40   | 0.13   | 0.06   | 0.06   | 1.40            |
> >
> >
> > ## **Paper Revision Updates**
> > We have submitted a revised version of our paper, with changes highlighted in blue. Below are the updates:
> >
> > (1)	Added the description of the recent paper "Closed-Loop Visuomotor Control with Generative Expectation for Robotic Manipulation" in the "Related Work" section of the manuscript.
> >
> > (2)	Added the results of the baseline algorithm SuSIE on the CALVIN ABC-D generalization task according to the suggestion of reviewer siH9 (Table 2 of the manuscript). Compared with all baseline algorithms, our proposed algorithm GEVRM achieves the highest performance.
> >
> > (3)	Added real-world experiments in the "Real-World Tasks" section of the manuscript appendix. The experiments show that the proposed method GEVRM can be effectively deployed to common real-world robot arm pick-and-place tasks.
> >
> > Again, we sincerely thank the reviewers for their constructive feedback. We believe all comments have been addressed in this revision but are happy to address any further comments from the reviewers.

---

### Author Response · Authors · 2024-11-28
**Friendly Reminder Regarding Upcoming Rebuttal Deadline**

Dear reviewers, I hope this message finds you well. This is a gentle reminder regarding the review of our manuscript.
We deeply appreciate the invaluable comments and feedback provided by reviewers. They are instrumental in enhancing the quality of our research. As per the schedule, the rebuttal phase is drawing to a close. We understand that you have a demanding schedule and a multitude of responsibilities, but we are keen to receive your feedback before the deadline. This will afford us the opportunity to address any questions or concerns you may have raised in a timely manner. We are eager to incorporate your insights to refine our work and would be grateful if you could share your thoughts prior to the rebuttal deadline.

Thank you very much for your hard work and support. Your dedication to the review process is greatly appreciated.

---

### Meta-Review · Area_Chair_fsHb · 2024-12-25

**Metareview:**

The paper introduces GEVRM, a closed-loop vision-language-action (VLA) model based on the internal model control (IMC) principle, designed to enhance robustness to external perturbations through the integration of text-guided video generation and prototype contrastive learning. The model achieves state-of-the-art performance on both standard and perturbed CALVIN benchmarks and demonstrates strong results in realistic robot tasks.

The reviewers unanimously acknowledged the paper's contributions, noting its (1) clear and well-motivated approach, (2) extensive experimental results demonstrating advantages over existing methods, (3) valuable insights from evaluations under external environmental disturbances, and (4) clear and well-structured presentation.

During the Author-Reviewer Discussion phase, the authors provided detailed responses that successfully addressed many of the reviewers' concerns, leading to score increases from multiple reviewers. With all reviewers in unanimous agreement to accept the paper, the AC recommends acceptance while encouraging the authors to carefully address both pre- and post-rebuttal comments to further strengthen the final version.

**Additional Comments On Reviewer Discussion:**

Since the reviewers were in unanimous agreement to accept this paper, no significant discussion took place during the Reviewer Discussion phase.

---

### Decision · Program_Chairs · 2025-01-22

Accept (Poster)